# Errors in Nanoparticle Growth Rates Inferred from Measurements in Chemically Reacting Aerosol Systems

Chenxi Li[1] and Peter H. McMurry[1]

[1]Department of Mechanical Engineering, University of Minnesota, Minneapolis, MN, 55455, USA

*Correspondence to*: Chenxi Li (lixx3838@umn.edu)

**Abstract.** In systems where aerosols are being formed by chemical transformations, individual particles grow due to the addition of molecular species. Efforts to improve our understanding of particle growth often focus on attempts to reconcile observed growth rates with values calculated from models. However, because it is typically not possible to measure the growth rates of individual particles in chemically reacting systems, they must be inferred from measurements of aerosol properties such as size distributions, particle number concentrations, etc. This work discusses errors in growth rates obtained using methods that are commonly employed for analyzing atmospheric data. We analyze "data" obtained by simulating the formation of aerosols in a system where a single chemical species is formed at a constant rate, $R$. We show that the maximum overestimation error in measured growth rates occurs for collision-controlled nucleation in a single-component system in the absence of a pre-existing aerosol, wall losses, evaporation or dilution, as this leads to the highest concentrations of nucleated particles. Those high concentrations lead to high coagulation rates that cause the nucleation mode to grow faster than would be caused by vapor condensation alone. We also show that preexisting particles, when coupled with evaporation, can significantly decrease the concentration of nucleated particles. This can lead to decreased discrepancies between measured growth rate and true growth rate by reducing coagulation between nucleated particles. However, as particle sink processes get stronger, measured growth rates can potentially be lower than true particle growth rates. We briefly discuss nucleation scenarios where the observed growth rate approaches zero while the true growth rate does not.

## 1 Introduction

Aerosol systems undergo transformations by processes that include coagulation, convection, deposition on surfaces, source emissions, nucleation, growth, etc. The aerosol general dynamic equation (GDE) (Friedlander, 2000;Gelbard and Seinfeld, 1979, 1980) describes the time rate of change of size-dependent particle concentration and composition by such processes. Recent work has focused on understanding processes that affect growth rates of freshly nucleated atmospheric nanoparticles (Smith et al., 2008;Smith et al., 2010;Riipinen et al., 2012;Hodshire et al., 2016;Kontkanen et al., 2016;Tröstl et al., 2016).This is important because a particle's survival probability increases with growth rates (McMurry and Friedlander, 1979;Weber et al., 1997;Kerminen and Kulmala, 2002;Kuang et al., 2010). Nucleated particles are more likely to form cloud condensation nuclei and affect climate when survival probabilities are high.

Following established conventions long used in modeling aerosol dynamics (Friedlander, 2000;Gelbard and Seinfeld, 1979, 1980), we define the particle "growth rate" as the net rate of change in diameter of individual particles due to the addition or removal of molecular species. (If evaporation exceeds addition, the growth rate would be negative.) While most work to date has focused on condensation and evaporation, chemical processes such as acid-base reactions, organic salt formation, liquid phase reactions, and the accretion of two or more organic molecules to form a larger compound having lower volatility may also contribute to growth (McMurry and Wilson, 1982;Barsanti et al., 2009;Riipinen et al., 2012;Lehtipalo 2014). In a chemically reacting system, the total diameter growth rate, $GR$, is given by the sum of all such processes:

$$\frac{dd_p}{dt} = GR = GR_{condensation/evaporation} + GR_{acid-base\ reactions} + GR_{accretion} + GR_{other}. \tag{1}$$

The effect of growth on the aerosol distribution function is given by (Heisler and Friedlander, 1977):

$$\left.\frac{\partial n}{\partial t}\right|_{Growth} = -\frac{\partial}{\partial d_p}\left[n(d_p, t)\frac{dd_p}{dt}\right], \tag{2}$$

where the aerosol number distribution, $n(d_p, t)$ is defined such that the number concentration of particles between $d_p$ and $d_p + dd_p$ is equal to $n(d_p, t)dd_p$. Coagulation, including the coagulation of a molecular cluster with a larger particle, can also lead to particle growth. It is worthwhile, however, to treat coagulation and growth separately. The extent to which the coagulation of freshly nucleated molecular clusters contributes to measured growth rates can be accurately determined only if the entire number distribution down to clusters of size 2 is accurately measured. In the absence of such data, the contributions of cluster coagulation to growth could erroneously be attributed to vapor uptake. Coagulation is accounted for with the coagulation integrals in the GDE and is a relatively well understood process that can be described with reasonable confidence in models (Kürten et al., 2018;Chan and Mozurkewich, 2001). Growth involves processes that are not well understood for chemically complex aerosol systems, such as the atmosphere (Barsanti et al., 2009;Riipinen et al., 2012;Hodshire et al., 2016).

Progress towards understanding growth can be achieved through efforts to reconcile GRs that are observed experimentally with values predicted by models. Such work requires that size- and time-dependent GRs be accurately determined from observations. The literature includes many reports of observed GRs (Stolzenburg et al., 2005;Wang et al., 2013;Riccobono, 2012;Tröstl et al., 2016), but uncertainties in reported values are typically not well understood.

Because it is usually not possible to measure the growth of individual particles as they undergo chemical
transformations, GRs are calculated indirectly using time-dependent observations of aerosol properties such as number
distributions or number concentrations larger than a given size. Those properties are typically affected by many
processes, some poorly understood, that can affect reported GRs to an unknown extent.
A variety of approaches have been used to extract GRs from observations. We refer to these values as $GR_m$, where the
subscript 'm' designates 'measured'.  Methods that we discuss include:
1. *Maximum Concentration Method (Kulmala et al., 2012)*. During a nucleation event, particle concentrations in
a given size bin increase from their initial values, passing through a peak before they eventually decrease. This
technique involves noting the times that this maximum occurred in different size bins. The growth rate is
obtained by first fitting a linear function of particle diameter (corresponding to the size bins) vs. time, and then
calculating the slope of the fitted function.
2. *Appearance Time Method (Lehtipalo 2014)*. This approach has been used  to analyze data from condensation
particle counter (CPC) batteries (Riccobono, 2014), particle size magnifier (PSM) (Lehtipalo 2014), etc.. In
brief, $GR_m$ is determined by the differences in concentration rise times (typically, either 5% or 50% of the
maximum) measured by  the instruments with differing minimum detection sizes. A variation of this approach
was reported by Weber et al. (1997), who estimated growth rates from the observed time delay in measurements
of sulfuric acid vapor and particles measured with a condensation particle counter having  a minimum
detectable size of about 3 nm.
3. *Log-normal Distribution Function Method (Kulmala et al., 2012)*. Lognormal distributions are fit to the
growing mode of nucleated particles. $GR_m$ is defined as the growth rate of the geometric mean size of these
distributions.
While these methods do not account for the effects of coagulation on measured changes in particle size, the literature
includes approaches that explicitly account for such effects (Lehtinen et al., 2004;Verheggen and Mozurkewich,
2006;Kuang et al., 2012;Pichelstorfer et al., 2017). Other work has applied the above techniques after confirming that
coagulation has an insignificant effect for the analyzed data (Kulmala et al., 2012)  or explicitly accounting for the
effects of coagulation on $GR_m$ (Stolzenburg et al., 2005;Lehtipalo et al., 2016).
This paper assesses errors of using $GR_m$ calculated using techniques commonly employed in the literature to infer
particle growth rates. Our results are especially germane to *GR* of freshly nucleated particles ranging in size from
molecular clusters to about 40 nm. We use time-dependent distribution functions calculated numerically by McMurry
and Li (2017) as "data". The only process contributing to the addition or removal of molecular species in that work
(i.e., to particle "growth rates" as is defined above) are condensation and evaporation. Because we understand this
model system perfectly, $GR_{true}$ (i.e., the net growth rate due molecular exchange through condensation and evaporation)
can be calculated exactly. Errors in $GR_m$ due to coagulation, wall deposition, scavenging by preexisting particles, or
dilution, are given by the difference between $GR_{true}$ and $GR_m$. We do not examine errors associated with convection,
source emission, etc.
We are not the first to examine factors that cause $GR_m$ to differ from $GR_{true}$. For example, Kontkanen (2016) used
simulations to show that discrepancies between measured growth rate based on appearance time (AGR) and growth
rate based on irreversible vapor condensation (CGR) can be significant. (Note $GR_{true}$ used in this paper differs from
CGR in that $GR_{true}$ also incorporates evaporation.) Our approach, which uses the non-dimensional formulation
described by McMurry and Li (2017), provides results that are generally applicable to nucleation and growth of a
single chemical species, so long as it is being produced by chemical transformations at a constant rate, $R$. We show
that the upper limit for overestimation of $GR_{true}$ by $GR_m$ occurs when nucleation takes place in the absence of pre-
existing aerosols and is collision-controlled (i.e., when evaporation rates from even the smallest clusters occur at rates
that are negligible relative to vapor condensation rates). Collision-controlled nucleation is an important limiting case
because there is growing evidence that atmospheric nucleation of sulfuric acid with stabilizing species is well-
described as a collision-controlled process (Almeida et al., 2013;Kürten et al., 2018;McMurry, 1980). Because cluster
evaporation, scavenging by preexisting aerosol, etc., all diminish the number of particles formed by nucleation,
overestimation of $GR_{true}$ due to coagulation decreases as these processes gain in prominence.  We do not explicitly
study the effect of growth by processes other than condensation or evaporation, such as heterogeneous growth
pathways that take place on or within existing particles. If such processes were to contribute significantly to growth,
they would lead to higher growth rates and therefore smaller relative errors in $GR_m$ due to coagulation. Additionally,
we point out when particle sink processes consume nucleated particles at a fast rate (e.g. strong effects of dilution or
scavenging by preexisting particles), $GR_m$ may not be used to estimate $GR_{true}$. Our results help to inform estimates of
uncertainties for systems with a single condensing species, or systems that can be modeled in a similar way to a single
species system (Kürten et al., 2018).

## 113   2 Methods

### 114   2. 1 Discrete-sectional model

We utilize the dimensionless discrete-sectional model described by McMurry and Li (2017) to simulate evolution of
particle size distribution for a system with a single condensing species. We assume that the condensing species is
produced at a constant rate by gas phase reaction. Our code uses two hundred discrete bins and 250 sectional bins,
with a geometric volume amplification factor of 1.0718 for neighboring sections.
Physical processes that affect particle growth, including wall deposition, loss to pre-existing particles, cluster
evaporation and dilution, can be characterized by dimensionless parameters in this model. In the present study,
however, not all aforementioned processes are discussed. Our previous work shows that wall losses, scavenging by
preexisting particles and dilution have qualitatively similar effects on aerosol dynamics. Therefore, in this work we
focus on preexisting aerosols and dilution to illustrate factors that contribute to errors in measured growth rates, and
do not explicitly discuss wall deposition. A single dimensionless parameter, $\sqrt{L}$ , is used to indicate the abundance of
preexisting particles, with larger $\sqrt{L}$ representing higher concentration of preexisting particles (or, equivalently, a
slower rate at which the nucleating species is produced by chemical reaction). $\sqrt{L}$ is calculated with the equation
$$\sqrt{L} = \frac{\frac{1}{4}\left(\frac{8k_bT}{\pi m_1}\right)^{1/2} A_{Fuchs}}{\sqrt{R\beta_{fm\ 11}}},$$ (3)
where $A_{Fuchs}$ is the Fuchs surface area concentration (Fuchs and Sutugin, 1971), $k_b$ is the Boltzmann constant, $m_1$ is
the mass of the monomer, $R$ is the condensing species production rate, $\beta_{11\ fm}$ is the monomer collision frequency
function. The loss rate for particles containing $k$ monomers is $\sqrt{L}/k^{1/2}$. This size dependence is included when
solving the coupled differential equations for time-dependent cluster concentrations. Similarly, the dimensionless
quantity $M$ that characterizes dilution is given by the expression
$$M = \frac{Q_{dil}/V}{\sqrt{R\beta_{fm\ 11}}},$$ (4)
where $Q_{dil}$ is the dilution flow rate and $V$ is the volume of the system. Note the fractional dilution loss is independent
of particle size. In addition to loss to pre-existing particles and dilution, we consider the effect of cluster evaporation
on particle growth with the assumption that evaporation follows the classical liquid droplet model. Two dimensionless
parameters, $E$ and $\Omega$, are needed to fully describe the evaporation process. The dimensionless evaporation parameter,
$E$ , is proportional to the saturation vapor concentration of the nucleating species, while $\Omega$ is the dimensionless surface
tension (Rao and McMurry, 1989;McMurry and Li, 2017). The evaporation rate for particles containing $k$ monomers,
$E_k$ , is calculated with a discretized equation of the form:
$$E_k = E c_{1k} \exp\left[\frac{3}{2}\Omega\left(k^{\frac{2}{3}} - (k-1)^{\frac{2}{3}}\right)\right],$$ (5)
where $c(i,k)$ is the dimensionless collision frequency between a monomer and a particle containing $k$ monomers. To
simplify our discussion, $\Omega$ is fixed to be 16 throughout this work (a representative value for the surface tension of
sulfuric acid aqueous solutions), while the value of $E$ is varied.
The solution to the GDE for a constant rate system ($R$=constant) depends on dimensionless time, cluster size and the
dimensionless variables $\sqrt{L}$, $M$, $E$, $\Omega$, etc., but is independent of the rate at which condensing vapor is produced by
chemical reaction. That rate is required to transform the computed nondimensional solutions to dimensional results
using simple multiplicative expressions given by McMurry and Li (2017):
$$N_k = \left(\frac{R}{\beta_{11\ fm}}\right)^{1/2} \widetilde{N}_k \ ; \ t = \left(\frac{1}{R\beta_{11\ fm}}\right)^{1/2} \tau \ ; \ d_p = \left(v_1^{1/3}\right)\tilde{d}_p.$$ (6)
In the above equations, $\widetilde{N}_k$ is the dimensionless concentration of particle containing $k$ monomers, $\tau$ is the
dimensionless time, $\tilde{d}_p$ is the dimensionless particle size and $v_1$ is the monomer volume. Assuming a monomer
volume of $1.62\times10^{-22}$ cm$^3$ (volume of one sulfuric acid plus one dimethylamine molecule with a density of
1.47g/cm$^3$), $\tilde{d}_p = 30$ would be equivalent to a dimensional particle size of 16.4 nm.
**2.2 Evaluation of measured growth rate ($GR_m$)**
At time $t_1$ and $t_2$, if two particle sizes $d_{p,t1}$ and $d_{p,t2}$ are used to represent the particle size distribution, the 'measured'
growth rate can be calculated using the following equation as a first order approximation
$$GR_m\left(\frac{d_{p,t_1}+d_{p,t_2}}{2}, \frac{t_2+t_1}{2}\right) = \frac{d_{p,t2}-d_{p,t1}}{t_2-t_1}. \tag{7}$$
If $d_{p,t_i}$ is available for a time series $\{t_i\}_{i=1,2,...}$, growth rate can also be obtained by derivatizing a fitting function
$d_p = d_p(t)$ to obtain growth rate at any time $t_a$:
$$GR_m\left(d_p, t_a\right) = \left.\frac{dd_p(t)}{dt}\right|_{t=t_a}. \tag{8}$$
To implement Eq. (7) or (8), it is necessary to choose a particle size that is representative of the particle size distribution
at a given time. The choice of this representative size varies among publications and can depend on the types of
available data. Based on previous studies (Kulmala et al., 2012;Lehtipalo 2014;Stolzenburg et al., 2005;Yli-Juuti,
2011), we have selected four representative sizes for discussion: $\tilde{d}_{p,mode}$, $\tilde{d}_{p,sr100}$, $\tilde{d}_{p,sr50}$ and $\tilde{d}_{p,tot50}$. At a given
time $\tau$, $\tilde{d}_{p,mode}$ is the particle size at which $d\tilde{N}(\tau)/dlog_{10}\tilde{d}_p$ reaches its local maximum. If the shape of the mode is
log-normal, $\tilde{d}_{p,mode}$ is equal to the geometric mean of the distribution. As suggested by Kulmala et al. (Kulmala et
al., 2012), the 'log-normal distribution method' involves calculating growth rates from observed time-dependent
trends of $\tilde{d}_{p,mode}$ . The 'maximum concentration method'  is based on the time when particles in a given size bin,
$\tilde{d}_{p,sr100}$ , pass through their maximum (100%) concentration (Lehtinen and Kulmala, 2003).  The 'appearance time'
method is based on the time when particle concentrations in a bin, $\tilde{d}_{p,sr50}$, pass through a specified percentage of its
maximum (we have used 50%). Growth rates are sometimes based on total concentrations of particles larger than a
specified size. We refer to the particle size above which the total number concentration of particles reaches 50% of its
maximum value as $\tilde{d}_{p,tot50}$. This approach is especially useful when measurements are carried out with a battery of
CPCs having differing cutoff sizes. For simplicity, in this paper we assume that CPC detection efficiencies increase
from 0% to 100% at a given cutoff size. In practice, measured size-dependent detection efficiencies are typically used
when analyzing CPC battery data. Figure 1 shows the location of these representative sizes at $\tau = 20, 60, 100$ for two
nucleation scenarios in the absence of preexisting particles. $\tilde{d}_{p,mode}$, $\tilde{d}_{p,sr100}$, $\tilde{d}_{p,sr50}$ and $\tilde{d}_{p,tot50}$ are marked as
points, with their y-coordinates representing particle concentrations at corresponding sizes.
As will be shown later, values of $GR_m$ obtained with $\tilde{d}_{p,mode}$ , $\tilde{d}_{p,sr100}$ , $\tilde{d}_{p,sr50}$ or $\tilde{d}_{p,tot50}$ are not equal. To
differentiate these cases, $GR_m$ are notated as $GR_{m,mode}$, $GR_{m,sr100}$, $GR_{m,sr50}$ and $GR_{m,tot50}$ accordingly.
**2.3 Evaluation of true growth rate ($GR_{true}$)**
The true growth rate ($GR_{true}$) defined in this paper follows the Lagrangian approach (Olenius et al., 2014), i.e. tracking
the volume change of individual particles, and only include molecular species exchange by condensation and
evaporation. It is calculated with the following expression:
$$GR_{true} = \frac{d\tilde{d}_p}{d\tau} = \frac{2}{\pi\tilde{d}_p^2}\frac{d\tilde{V}}{d\tau} = \frac{2}{\pi\tilde{d}_p^2} \cdot \frac{\tilde{V}+c(i,k)\tilde{N}_1 \cdot d\tau - E_k \cdot d\tau - \tilde{V}}{d\tau} = \frac{2(c(i,k)\tilde{N}_1 - E_k)}{\pi\tilde{d}_p^2}, \tag{9}$$
where $\tilde{d}_p$ is the representative size, $\tilde{N}_1$ is the concentration of monomers, and $E_k$ is the particle evaporation rate given
by Eq. (5).
If evaporation is negligible ($E = 0$) and $\tilde{N}_1$ is constant, Eq. (9) leads to a higher growth rate for smaller particles,
mainly because of the increased monomer collision frequency relative to particle size (Tröstl et al., 2016). Throughout
this work Eq. (9) is used to evaluate true particle growth rate. Note $GR_{true}$ is calculated from dimensionless size and
time, and is therefore dimensionless. Since we focus on relative values of true and measured growth rates, our
conclusions are unaffected by the dimensionality of *GR*. However, dimensionless growth rates can be converted to
dimensional values with Eq. (6).
**3. Results and discussion**
**3.1 Error of using *GR*$_{m,mode}$ as *GR*$_{true}$**
As mode diameter ($\tilde{d}_{p,mode}$) is often employed to derive particle growth rate, in this section we discuss the error of
using *GR*$_{m,mode}$ as a substitute for *GR*$_{true}$ in the absence of preexisting particles. The effect of preexisting particles is
discussed in Sect. 3.3.
Both condensation and coagulation lead to growth of $\tilde{d}_{p,mode}$. To understand their relative importance, we attribute
*GR*$_{m,mode}$ to three processes: monomer condensation minus evaporation (*GR*$_{true}$), coagulation of the mode with clusters
(*GR*$_{m,cluster}$) and self-coagulation of the mode (*GR*$_{m,self}$). The latter two processes are the main causes of the discrepancy
between *GR*$_{m,mode}$ and *GR*$_{true}$. To evaluate *GR*$_{m,cluster}$ and *GR*$_{m,self}$, the range of 'clusters' and 'mode' are defined as
illustrated in Fig. 1 by the two shaded regions at $\tau = 100$: clusters (beige) and nucleation mode (light blue). Clusters
and nucleation mode are separated by $\tilde{d}_{p,min}$, where $d\tilde{N}/dlog_{10}\tilde{d}_p$ is at a local minimum. Stolzenburg et al.(2005)
assumed the nucleation mode is lognormal and calculated *GR*$_{true}$ and *GR*$_{m,self}$ with the method of moments. In this
work, since the mode for collsion-controlled nucleation deviates significantly from log-normal (see Fig. 1a), no
assumption regarding the shape of the nucleation mode is made. Instead, *GR*$_{m,cluster}$, *GR*$_{m,self}$ are calculated with the
first order numerical approximation method outlined in Appendix A.
The calculation results are summarized by Fig. 2. We first consider collision-controlled nucleation (*E*=0). For this
nucleation scenario, Fig. 2a shows $\tilde{d}_{p,mode}$ on the left y axis and growth rate values on the right. A third order
polynomial is used for fitting $\tilde{d}_{p,mode} = \tilde{d}_{p,mode}(\tau)$ and is plotted as a solid black line. Differentiating the fitted
polynomial with respect to time gives the value of *GR*$_{m,mode}$. It is clear that *GR*$_{true}$ only accounts for a small fraction
(17%-20%) of *GR*$_m$ and is on par with contribution of *GR*$_{m,cluster}$ (15%-22%). Self-coagulation is the major contributor
(62%-78%) to *GR*$_m$. Thus, using *GR*$_{m,mode}$ as a substitute for *GR*$_{true}$ leads to an overestimation by as much as a factor
about 6. We believe collision-controlled nucleation (*E*=0) in the absence of other particle loss mechanisms such as
wall deposition (*W*=0) and scavenging by pre-existing particles ($\sqrt{L}$=0) provides an upper limit for overestimation of
*GR*$_{true}$ for a constant rate system (*R*=constant). This is because these conditions lead to the maximum number of
particles that can be produced by nucleation. High concentrations lead to high coagulation rates, and it is coagulation
that is primarily responsible for errors in $GR_m$. Furthermore, as is discussed below, the absence of evaporation and
scavenging by nucleated particles keeps monomer concentrations low relative to values achieved when $E{\neq}0$ (see Fig.
2a). Low monomer concentrations reduce the value of $GR_{true}$, thereby increasing relative errors in $GR_m$.
Distinctive features of particle growth emerge when cluster evaporation is included by setting $E = 1{\times}10^{-3}$. Figure
2b shows results for this nucleation scenario. Most noticeably, particles grow considerably faster at early stages of
simulation. This occurs because evaporation depletes clusters and correspondingly increases monomer concentration.
In the absence of pre-existing particles, monomer concentration accumulates until the supersaturation is high enough
for nucleation to take place (see figure 2c). The accumulated monomers then rapidly condense on the nucleated
particles, leading to the rapid particle growth shown in figure 2b. To capture this rapid growth, two third-order
polynomials are used to fit $\tilde{d}_{p,mode}$ values for $\tau < 40$ and $\tau > 35$ respectively, with an overlapping region for $35 <$
$\tau < 40$. Furthermore, in comparison to collision-controlled nucleation, contribution of $GR_{m,cluster}$ to $GR_{m,mode}$ becomes
negligible, due to decreased cluster concentration by evaporation. For $\tau > 30$, $GR_{true}$ accounts for about 40%-55% of
$GR_{m,mode}$, larger than that of collision-controlled nucleation; for $\tau < 25$, $GR_{true}$ almost entirely accounts for $GR_{m,mode}$
and even exceeds $GR_{m,mode}$ at the very beginning of the nucleation. $GR_{true}/GR_{m,mode} >1$ indicates a rapidly forming
nucleation mode, where freshly nucleated particles enter the mode and skew the mode distribution toward smaller
sizes, slowing down the shift of the mode peak towards larger values.
Increase of $GR_{true}/GR_{m,mode}$ by evaporation is explained by the elevated monomer concentration due to particle
volatility and the smaller number of particles formed by nucleation: the former increases $GR_{true}$, and the latter decreases
$GR_{m,self}$ and $GR_{m,cluster}$. Figure 2c plots monomer concentration $\widetilde{N}_1$ as a function of time for several values of $E$.
Noticeably, monomer concentration elevates with $E$ since higher cluster evaporation rates require higher monomer
concentrations (i.e., higher supersaturation) to overcome the energy barrier of nucleation. Once nucleation takes place,
high monomer concentration leads to rapid nanoparticle growth rates.
Figure 2d shows $GR_{true}/GR_{m,mode}$ at $\tau = 30, 50, 100, 150$ for several $E$ values. At a given time, $GR_{true}/GR_{m,mode}$ clearly
increases with $E$: when evaporation rates are not negligible (i.e., $E{\neq}0$), $GR_{m,mode}$ is closer to $GR_{true}$ than occurs when
$E$=0. Again, this is because the elevated monomer concentrations increase $GR_{true}$ and the lowered concentrations of
clusters and nucleated particles decrease $GR_{m,cluster}$ and $GR_{m,self}$. As $E$ approaches 0, the value of $GR_{true}/GR_{m,mode}$
converges to that of the collision-controlled nucleation (~0.2). One data point, corresponding to $E = 5{\times}10^{-3}$ and
$\tau = 30$, with a value of 1.8, is not shown in Fig. 2d. It has a value significantly greater than unity because of the large
quantities of nucleated particles entering the mode, skewing the mode peak toward smaller sizes.
**3.2 Comparison of representative sizes**
In this section we examine how observed growth rate depends on the choice of a representative size. The application
of $GR_{m,mode}$ to deduce $GR_{true}$, though convenient in practice, depends on the existence of a nucleation mode. However,
the nucleation mode is usually not well defined in the early stage of nucleation. In contrast, growth rate based on other
representative sizes ($\tilde{d}_{p,sr50}$ , $\tilde{d}_{p,sr100}$ and $\tilde{d}_{p,tot50}$) are not dependent on mode formation and are available for all
particle sizes. In light of this, $GR_{m,sr100}$ , $GR_{m,sr50}$, $GR_{m,tot50}$ have often been employed to describe the growth rate of
small particles (<5nm). The effects of pre-existing particles are neglected in this section (i.e., $\sqrt{L} = 0$) but are
discussed in Sect. 3.3.
For collision-controlled nucleation, $\tilde{d}_{p,mode}$ , $\tilde{d}_{p,sr50}$, $\tilde{d}_{p,sr100}$, $\tilde{d}_{p,tot50}$ are plotted as functions of time in Fig. 3a. The
magnitude of the representative sizes follow $\tilde{d}_{p,mode}$< $\tilde{d}_{p,bin100}$< $\tilde{d}_{p,tot50}$< $\tilde{d}_{p,bin50}$, as was previously illustrated in
Fig. 1a. $\tilde{d}_{p,mode}$< $\tilde{d}_{p,bin100}$ indicates that a certain measurement bin first reaches its maximum concentration and
becomes a local maximum at a later time. This is true for collision-controlled nucleation with a decreasing peak
concentration but is not necessarily true for other nucleation scenarios. The observed growth rate (i.e. slope of curves
in Fig. 3a) are shown in Fig. 3b as a function of representative size, with a clear relationship $GR_{m,mode}$ <$GR_{m,sr100}$
<$GR_{m,tot50}$<$GR_{m,sr50}$. Note that $GR_{m,mode}$ is not available for small sizes, indicating the nucleation mode is yet to form
at the early stage of nucleation. Figure 3c shows $GR_{true}/GR_m$ as a function of representative size, with $GR_{true}$ calculated
with Eq. (9). Clearly $GR_{true}$ accounts for the highest percentage of $GR_m$ at the start of nucleation. This is partly due
to higher monomer concentrations (see red solid curve in Fig. 2c) and partly due to Eq. (9) that leads to higher true
growth rate for smaller particles: the addition of a monomer leads to a bigger absolute as well as fractional diameter
growth for small particles.
Figure 3d-3f are counterparts of Fig. 3a-3c, but with evaporation constant $E$ set to $1\times10^{-3}$. Figure 3d show that $\tilde{d}_{p,sr50}$
and $\tilde{d}_{p,tot50}$ increase relatively slowly at the start of the simulation (see the amplified figure at the lower right corner
of Fig. 3d; for reference, the dimensionless sizes of monomer, dimer and trimer are 1.24, 1.56 and 1.79 respectively).
Subsequently, a marked change slope of the $\tilde{d}_p = \tilde{d}_p(\tau)$ curve is observed, indicating accelerated particle growth.
This reflects that nucleation occurs with a burst of particle formation following a process of monomer and cluster
accumulation. The slow growth of the smallest clusters is an indication that the accumulation process is slow due to
the strength of the Kelvin effect.
Figure 3e shows $GR_m$ obtained by curve fitting after the nucleation burst and Fig. 3f shows the corresponding
$GR_{true}/GR_m$ values. Different from collision-controlled nucleation, there is a sharp rise of $GR_{true}/GR_m$ value at the start
of nucleation. This is due to the sharp decrease of the evaporation term in Eq. (9), causing the value of $GR_{true}$ to
increase sharply. As nucleation progresses, the ratio of $GR_{true}$ to $GR_{m,sr100}$, $GR_{m,tot50}$ and $GR_{m,sr50}$ comes close to 1,
with $GR_{m,mode}$ not yet available. Eventually, $GR_{true}/GR_m$ for all representative sizes decreases and fall into the range
of 30%-50%, with $GR_m^{mode}$ giving the best estimate of $GR_{true}$. Note the value of $GR_{true}/GR_{m,mode}$ significantly
exceeds unity for $\tilde{d}_p \in [5,11]$ due to the distortion of the mode toward smaller sizes by high flux of freshly nucleated
particles into the mode.
**3.3 Effect of pre-existing particles**
Pre-existing particles act as particle sinks to decrease the intensity of nucleation. Similarly, in chamber experiments,
though loss to pre-existing particles is often eliminated by using air that is initially particle-free, loss of particles to
chamber walls is inevitable. Since wall loss and loss to preexisting particles have qualitatively similar effect on
nucleation (McMurry and Li, 2017), we selectively examine the effect of preexisting particles on growth rate
measurements to qualitatively illustrate the effects of all of these processes. To probe the initial stage of nucleation,
we use $\tilde{d}_{p,bin50}$ as the basis for our analysis, with a comparison of representative sizes presented at the end of this
section. As to the magnitude of $\sqrt{L}$, we choose $\sqrt{L} \in [0,0.3]$ based on previous work. It was shown in Fig. 2b in
McMurry and Li (2017) that as $\sqrt{L}$ exceeds 0.1, particle size distributions begin to deviate discernably from the
collision-controlled case. In addition, $\sqrt{L} \approx 0.2$ was observed in the ANARChE field campaign carried out in Atlanta
for nucleation events with sulfuric acid as the major nucleating species (Kuang et al., 2010).
The influence of preexisting particles on the discrepancy between true and measured growth rate ($GR_{true}/GR_m$) is
twofold. On one hand, preexisting particles can decrease monomer concentration which leads to a smaller $GR_{true}$. On
the other hand, preexisting particles reduce coagulation by scavenging nucleated particles, which could result in a
narrower gap between $GR_{true}$ and $GR_m$. Therefore, the response of $GR_{true}/GR_m$ to $\sqrt{L}$ depends on the relative magnitude
of these two competing effects. Figure 4a shows $\tilde{d}_{p,sr50}$ as a function of time for several $\sqrt{L}$ values and Fig. 4b displays
the corresponding $GR_{true}/GR_m$ values. It can be seen that $GR_{true}/GR_m$ positively correlates with $\sqrt{L}$, indicating
preexisting particles are more effective in removing nucleated particles than reducing monomer concentrations. In
fact, as further demonstrated by Fig. 4c, monomer concentrations (leftmost point of all the curves) are barely affected:
scavenging of monomers by preexisting particles are offset by less condensation of monomers onto nucleated particles.
Note that for the range of $\sqrt{L}$ values examined, the presence of preexisting particles alter $GR_{true}/GR_m$ values by no
more than 50% for collision-controlled nucleation.
Figures 4d-4f show the same quantities as are shown in Fig. 4a-4c, but with $E$ set to $1\times10^{-3}$ instead of zero. In
contrast to collision-controlled nucleation, pre-existing particles significantly affect the nucleation process when
cluster evaporation is taken into account. As $\sqrt{L}$ increases, Fig. 4e shows $GR_{true}/GR_m$ converges to a value slightly
larger than unity. This indicates that the contribution of coagulation to measured growth rate approaches zero as $\sqrt{L}$
becomes large; or equivalently, the concentration of nucleated particles is severely decreased by pre-existing particles.
Values of $GR_{true}/GR_{m,sr50}$ slightly exceed unity for large sizes (Fig. 4f) due to the slightly higher condensational growth
rates of smaller particles in the nucleation mode. This shifts values of $\tilde{d}_{p,sr50}$ towards smaller sizes than would occur
if all particles were to grow at the same rate, causing $GR_{m,sr50}$ to be smaller than $GR_{true}$.
The decrease of nucleated particle concentration is further demonstrated in Fig. 4f. From $\sqrt{L} = 0$ to $\sqrt{L} = 0.3$, the
peak concentration of nucleated particles dropped by about three orders of magnitude. Such a decrease in concentration
of nucleated particles results from the limiting effect of $\sqrt{L}$ on monomer concentration. If pre-existing particles are
absent, then no major loss mechanisms for monomers exist prior to the nucleation burst. Monomer would accumulate
until the nucleation energy barrier can be overcome: the higher the energy barrier, the higher the monomer
concentration prior to nucleation, as shown in Fig. 2c. The elevated monomer concentration then leads to rapid growth
of freshly nucleated particles immediately following the nucleation burst. However, in the presence of pre-existing
particles (i.e., $\sqrt{L} \neq 0$), monomer concentration can only increase to the point where its production and consumption
by preexisting particles reach balance, prohibiting its concentration from reaching a high value even prior to the
nucleation burst. To facilitate comparison with experimental results, in Appendix B we provide an example of
conversion from dimensionless distributions and growth rates to dimensional ones.
Finally, Fig. 5 examines the difference between representative sizes used to calculate $GR_m$ when loss to preexisting
particles is accounted for. Two cases are presented: (1) collision-controlled nucleation ($E$=0) with $\sqrt{L} = 0.2$ (Fig. 5a-
5c) and (2) nucleation accounting for both cluster evaporation and scavenging by preexisting particles ($E =$
$1\times10^{-3}$ and $\sqrt{L} = 0.2$; Fig. 5d-5f). For collision-controlled nucleation with $\sqrt{L} = 0.2$, the preexisting particles
changes nucleation only slightly, although $GR_m$ decreases and $GR_{true}/GR_m$ increases both to a minor extent compared
to collision-controlled nucleation in the absence of a preexisting aerosol (compare Fig. 5a-5c to Fig. 3a-3c). The
analysis made in the discussion of Fig. 3a-3c still stands for Fig. 5a-5c. For nucleation with evaporation and preexisting
particles coupled together (Fig. 5d-5f), three features are worthy of attention. Firstly, compared to evaporation-only
nucleation, $GR_m$ is significantly decreased for small particle sizes. For $\tilde{d}_p < 10$, $GR_m$ is no larger than 0.7 with
preexisting particles but can be greater than 1.5 without (refer to Fig. 3e). Secondly, as shown in Fig. 5f, $GR_{true}/GR_{m,sr50}$
and $GR_{true}/GR_{m,tot50}$ come close to unity due to negligible coagulation effects. Third, $GR_{true}/GR_{m,mode}$ is between 1.2 and
1.5 and $GR_{true}/GR_{m,sr100}$ is between 1.1 and 1.2 for $\tilde{d}_p > 10$, indicating the true growth will be slightly underestimated
if $\tilde{d}_{p,mode}$ or $\tilde{d}_{p,sr100}$ are used to infer $GR_{true}$.

### 3.4 Underestimation of $GR_{true}$

In previous sections, mainly overestimation of the $GR_{true}$ by measured growth rate, $GR_m$, has been discussed. Though
we do no quantitatively study underestimation of $GR_{true}$ by $GR_m$, in this section we show that in a constant rate system
where particle sink processes (i.e. dilution and loss to pre-existing particles) strongly decrease the concentration of
nucleated particles, $GR_m$ can approach zero and cannot be utilized to estimate $GR_{true}$. Figure 6 shows such nucleation
scenarios for (a) collision-controlled nucleation with M = 0.1 and (b) collision-controlled nucleation with $\sqrt{L} = 1.5$.
In both cases other sink processes were set equal to zero. As shown in both Fig. 6a and 6b, particle size distributions
approach steady state after $\tau = 100$. As a result, the measured growth rate $GR_m$ approaches zero beyond $\tau = 100$. At
the same time, true condensational growth remains finite since monomer concentration remains steady state after $\tau =$
20. Therefore, other methods have to be utilized to infer $GR_{true}$ in such situations.

### 4 Conclusions

We used a discrete-sectional model to solve a dimensionless form of aerosol population balance equation for a single-
species system. True growth rate and various "measured" growth rates were examined for a variety of nucleation
scenarios. Based on the simulation results, we draw the following conclusions:
1.   Simulated data shows that for collision-controlled nucleation without preexisting particles, growth rates

353         inferred from the modal size of nucleated particles ($GR_{m,mode}$) is as much as 6 times greater than true growth

354         rates due to vapor condensation ($GR_{true}$).

2. In the absence of preexisting particles or other sink processes, comparison of different growth rates based on different representative sizes indicates the relationship $GR_{m,mode}<GR_{m,sr100}<GR_{m,tot50}<GR_{m,sr50}$ holds true for collision-controlled nucleation. If clusters evaporate, the nucleation process is characterized by rapid particle growth following the nucleation burst.

3. Both evaporation and scavenging by preexisting particles can reduce the concentration of particles formed by nucleation. Lower particle concentrations reduce the effect of coagulation on $GR_m$, so overestimation of $GR_{true}$ by $GR_m$ is lower than is found in the absence of these processes.

4. Preexisting particles have dramatically different effects on collision-controlled nucleation and nucleation with cluster evaporation. For $\sqrt{L} \in [0,0.3]$, collision-controlled nucleation is only slightly affected. However, if preexisting particles are coupled with evaporation, the number of nucleated particles can drop significantly, thus reducing the contribution of coagulation to measure growth rates.

5. $GR_m$ can underestimate $GR_{true}$ in a system with strong dilution or other particle sink processes. Particle size distributions in such nucleation scenarios can approach a steady state that leads to a $GR_m$ close to 0, which underestimates $GR_{true}$.

**Appendix A**

To evaluate the contribution of self-coagulation of the mode ($GR_{m,self}$) and cluster coagulation ($GR_{m,clsuter}$) to
measured growth rate based on mode diameter ($GR_{m,mode}$), we used the following first order numerical approximation
method:
1.  Find particle size distribution $\tilde{n} = \tilde{n}(k,\tau)$ at a given time $\tau$. $k$ is the number of monomers in a particle and $\tilde{n}_k$
is the concentration of particles that contains $k$ molecules. Since the simulation code only reports discrete particle
concentration for each bin, an interpolation is performed using Matlab function *griddedInterpolant.m*.
2.  Find the value $k = k_{max}$ at which $3\log(10)\,k\tilde{n}(k,\tau)$ is locally maximized. A prefactor $3\log(10)\,k$ is
multiplied to $\tilde{n}(k,\tau)$ to convert the particle size distribution to $d\tilde{N}/dlog_{10}\tilde{d}_p$. The mode diameter is then given
by $\tilde{d}_{p,mode}(\tau) = \left(\frac{6k_{max}}{\pi}\right)^{1/3}$
3.  Use the following integration equations to obtain number distribution of the mode at time $\tau + \Delta\tau$ assuming only
one process causes the distribution to shift.
For self-coagulation:
$\tilde{n}_{self}(k,\tau+\Delta\tau) = \tilde{n}(k) + 0.5*\Delta\tau*\int_L^k c(x,k-x)\tilde{n}(x,\tau)\tilde{n}(k-x,\tau)dx - \int_L^H c(x,k)\tilde{n}(k,\tau)\tilde{n}(x,\tau)dx.$  (A1)
For coagulation with clusters:
$\tilde{n}_{cluster}(k,\tau+\Delta\tau) = \tilde{n}(k,\tau) + 0.5\cdot\Delta\tau\cdot\int_{L_c}^{H_c} c(x,k-x)\tilde{n}(x,\tau)\tilde{n}(k-x,\tau)H(H_c-k+x)dx + \Delta\tau\cdot$
$\int_{L_c}^{H_c} c(x,k-x)\tilde{n}(x,\tau)\tilde{n}(k-x,\tau)H(k-x-H_c)dx - \Delta\tau\cdot\int_{L_c}^{H_c} c(x,k)\tilde{n}(x,\tau)\tilde{n}(k,\tau)dx.$  (A2)
In the above equations, $L$ and $H$ are the lower and upper boundary of the mode, $L_c$ and $H_c$ are the lower and
upper boundary of clusters, $c(i,j)$ is the collision frequency function, $H(x)$ is the Heaviside step function. $\Delta\tau$ is
typically set between 0.1 to 1.
4.  Find the $k$ values at which $3\log(10)\,k\tilde{n}_{self}(k,\tau+\Delta\tau)$ and $3\log(10)\,k\tilde{n}_{cluster}(k,\tau+\Delta\tau)$ are locally
maximized. The corresponding diameters are $\tilde{d}_{p,self}(\tau+\Delta\tau)$ and $\tilde{d}_{p,cluster}(\tau+\Delta\tau)$.
5.  The growth rate due to self-coagulation and coagulation with clusters are then given by
$GR_{m,self} = \frac{\tilde{d}_{p,self}(\tau+\Delta\tau)-\tilde{d}_{p,mode}(\tau)}{\Delta\tau};\ GR_{m,cluster} = \frac{\tilde{d}_{p,cluster}(\tau+\Delta\tau)-\tilde{d}_{p,mode}(\tau)}{\Delta\tau}.$  (A3)

**Appendix B**


To facilitate comparison between dimensionless simulation results and experimental results, or previous dimensional
simulation results, we convert selected dimensionless simulation results to dimensional quantities using Eq. (6).
Specifically, we assume the monomer production rate is $R = 1 \times 10^6$ cm$^{-3}$ s$^{-1}$ and the monomer has a volume of
$1.62 \times 10^{-22}$ cm$^3$ and a density of $1.47$ g cm$^{-3}$. The collision frequency function for monomers, $\beta_{11\,fm}$, is
$4.27 \times 10^{-10}$ cm$^3$ s$^{-1}$, calculated at atmospheric pressure and 300 K. We consider two nucleation scenarios. The first
is collision-controlled nucleation in the presence of pre-existing particles, with $\sqrt{L}$ set to 0.2. The second scenario is
nucleation with evaporation in the presence of pre-existing particles. The evaporation constant in this case is $E =$
$1 \times 10^{-3}$ and $\sqrt{L}$ is 0.2. Both these cases are discussed in Sect. 3.3. The converted dimensional results are shown in
Fig. B1, with relevant dimensional quantities displayed in the figure.
**Acknowledgements**
This research was supported by the US Department of Energy's Atmospheric System Research, an Office of Science,
Office of Biological and Environmental Research program, under grant number DE-SC0011780.
**Nomenclature**
Collision-controlled nucleation: a limiting case for nucleation where all collisions between condensing (nucleating)
vapor occur at the rate predicted by kinetic theory and particles stick with 100% efficiency. Vapor does not
subsequently evaporate from particle surfaces, nor are particles scavenged by pre-existing particles or the chamber
wall
$\tilde{d}_{p,min}$: particle size corresponding to the local minimum in a $d\widetilde{N}/dlog_{10}\tilde{d}_p$ representation of particle size distribution
$\tilde{d}_{p,mode}$: particle size corresponding to the local maximum in a $d\widetilde{N}/dlog_{10}\tilde{d}_p$ representation of particle size
distribution
$\tilde{d}_{p,sr50}$: particle size of a measurement bin where particle concentration reaches 50% of its maximum value
$\tilde{d}_{p,sr100}$: particle size of a measurement bin where particle concentration reaches maximum value
$\tilde{d}_{p,tot50}$: particle size above which total particle concentration reaches 50% of its maximum value
$GR_{m,mode}$: measured dimensionless growth rate based on $\tilde{d}_{p,mode}$
$GR_{m,sr50}$: measured dimensionless growth rate based on $\tilde{d}_{p,sr50}$
$GR_{m,sr100}$: measured dimensionless growth rate based on $\tilde{d}_{p,sr100}$
$GR_{m,tot50}$: measured dimensionless growth rate based on $\tilde{d}_{p,tot50}$
$GR_{true}$: true dimensionless particle growth rate attributed to the net flux of condensing vapors onto particle surface
(i.e., the condensation rate minus the evaporation rate)
$GR_{m,clsuter}$: measured dimensionless particle growth rate attributed to coagulation with clusters
$GR_{m,self}$: measured dimensionless growth rate attributed to self-coagulation of particles in the nucleation mode
$E, \Omega$: dimensionless parameters characterizing evaporation rates of particles, derived from the liquid droplet model.
$E$ can be regarded as a dimensionless form of saturation vapor pressure of the condensing molecules and $\Omega$ a
dimensionless form of surface tension. $\Omega$ assumes a constant value of 16in this work.
$\sqrt{L}$: dimensionless parameter characterizing fractional loss rate of monomer or nucleated particles to pre-existing
particles
$\widetilde{N}_k$: dimensionless concentration of particles containing $k$ monomers (i.e., k molecules of condensed vapor)

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

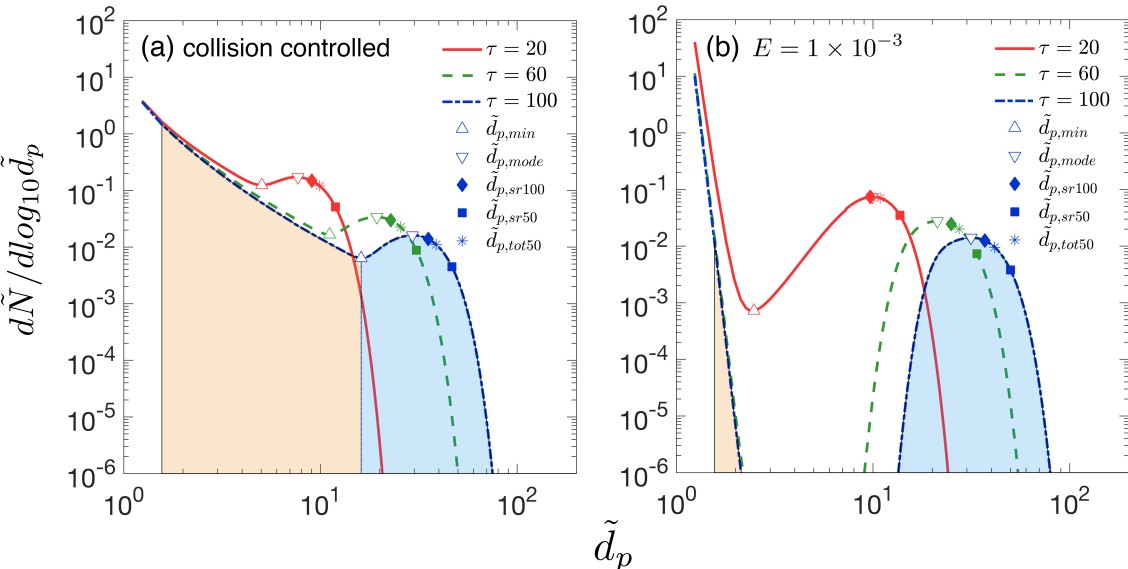

**Figure 1.** Particle size distributions at dimensionless times $\tau = 20, 60, 100$ **(a)** for collision-controlled nucleation
(*E=0*) and **(b)** when evaporation is included with $E = 1 \times 10^{-3}$. Division of the distribution into monomer, cluster
and nucleation mode is displayed for $\tau = 100$, with beige and light blue indicating the range of clusters and nucleation
mode. Clusters and nucleation mode are separated by $\tilde{d}_{p,min}$, where $d\tilde{N}/dlog_{10}\tilde{d}_p$ is at a local minimum.
Characteristic sizes $\tilde{d}_{p,mode}$, $\tilde{d}_{p,sr100}$, $\tilde{d}_{p,sr50}$ and $\tilde{d}_{p,tot50}$ are marked for each time. The relationship between
symbols and characteristic sizes is shown only for $\tau$=100.

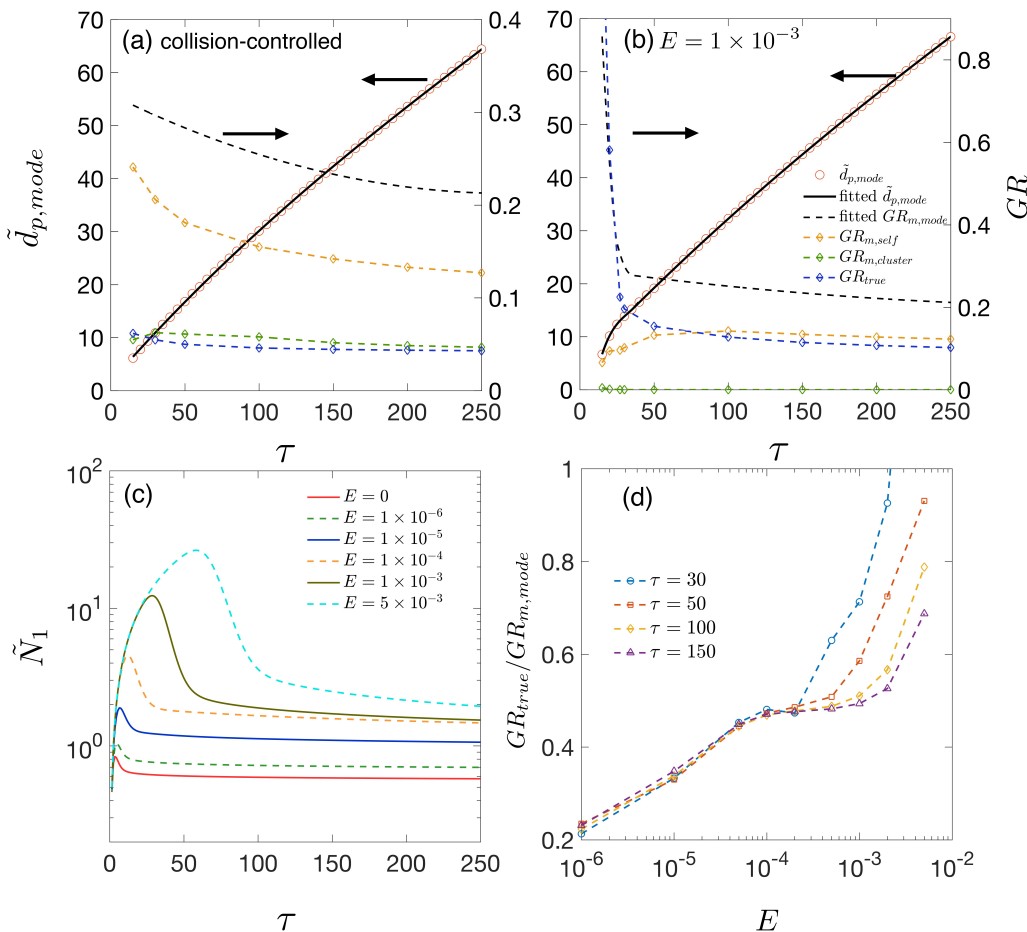


**Figure 2. (a)** $\tilde{d}_{p,mode}$ and various growth rates as functions of time for collision-controlled nucleation. Dashed black lines show the value of $GR_{m,mode}$. Yellow, green and blue dashed lines represent $GR_{m,self}$, $GR_{m,cluster}$ and $GR_{true}$ respectively. **(b)** The same quantities as are shown in (a) but with the evaporation constant set to $E = 1 \times 10^{-3}$. For both Fig. 2a and 2b, the left axis shows value for the solid lines and the right axis shows values for the dashed lines. **(c)** Monomer concentration as functions of time for different values of $E$. **(d)** $GR_{true}/GR_{m,mode}$ for different values of $E$ at $\tau = 30, 50, 100, 150$.


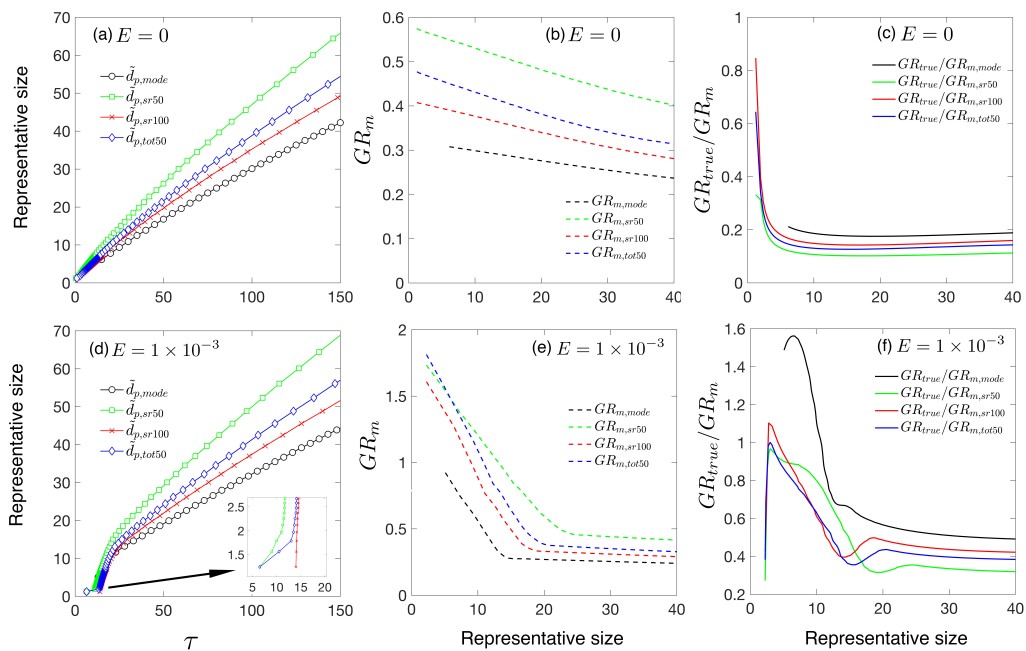


**Figure 3. (a)** $\tilde{d}_{p,mode}$, $\tilde{d}_{p,sr100}$, $\tilde{d}_{p,tot50}$, $\tilde{d}_{p,bin50}$ as functions of time. **(b)** Measured growth rates $GR_{m,mode}$, $GR_{m,sr50}$,
$GR_{m,sr100}$, $GR_{m,tot50}$ as functions of representative sizes. **(c)** Ratio of true growth rate to measured growth rate,
$GR_{true}/GR_m$. Figures 3a-3c are for collision-controlled nucleation with $E$=0. Figures 3d-3f show the same quantities
as are shown in Fig. 3a-3c but with $E = 1 \times 10^{-3}$.

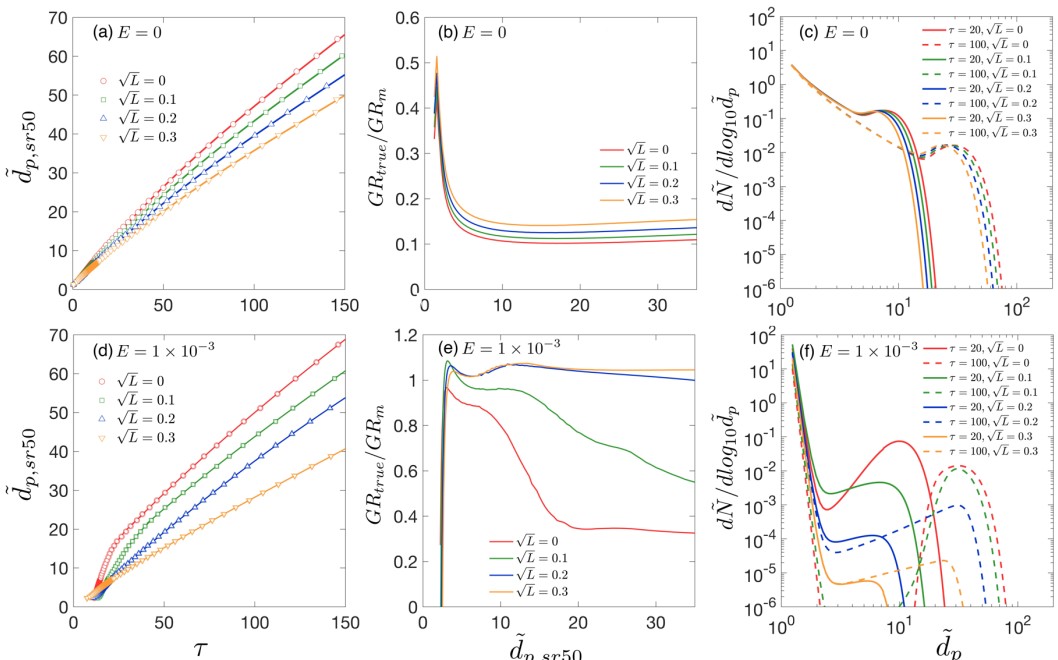

**Figure 4.** Effect of preexisting particles on particle growth rate. **(a)** $\tilde{d}_{p,sr50}$ as a function of time. **(b)** Ratio of true growth rate to measured growth rate, $GR_{true}/GR_{m,sr50}$. **(c)** Particle size distributions at $\tau = 20$ and $\tau = 100$. Figures 4a-4c are for collision-controlled nucleation with $E = 0$ and $\sqrt{L} = 0, 0.1, 0.2, 0.3$. Figures 4c-4d show the same quantities as are shown in Fig. 4a-4c but with $E = 1 \times 10^{-3}$.

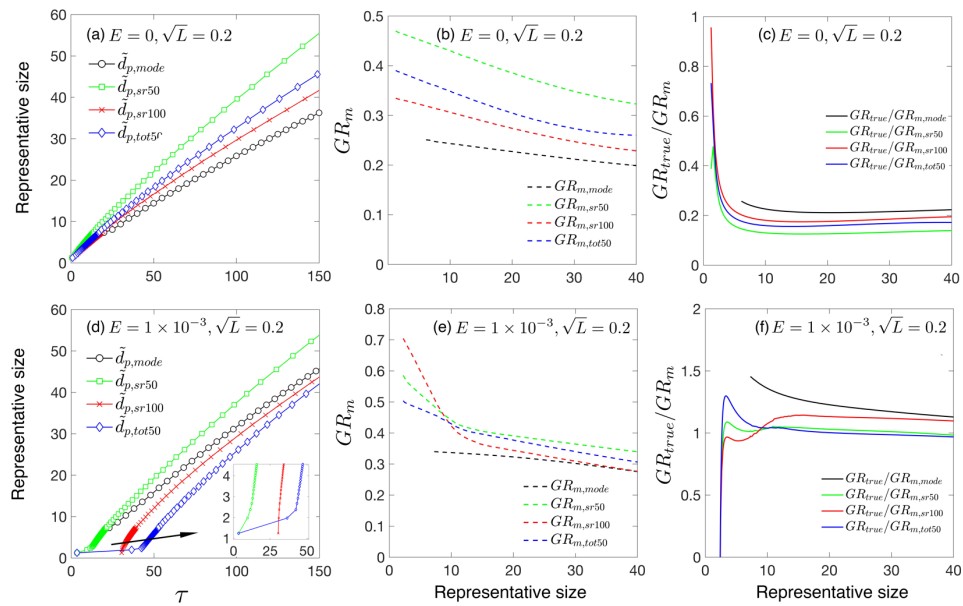

596

**Figure 5. (a)** $\tilde{d}_{p,mode}$, $\tilde{d}_{p,sr100}$, $\tilde{d}_{p,tot50}$, $\tilde{d}_{p,bin50}$ as functions of time. **(b)** Measured growth rate $GR_{m,mode}$ , $GR_{m,sr50}$, $GR_{m,sr100}$, $GR_{m,tot50}$ as functions of representative sizes. **(c)** Ratio of true growth rate to measured growth rate, $GR_{true}/GR_m$. Figures 5a-5c are for collision-controlled nucleation with $E = 0$ and $\sqrt{L} = 0.2$. Figures 5d-5f show the same quantities as are shown in Fig. 5a-5c but with $E = 1 \times 10^{-3}$.

601

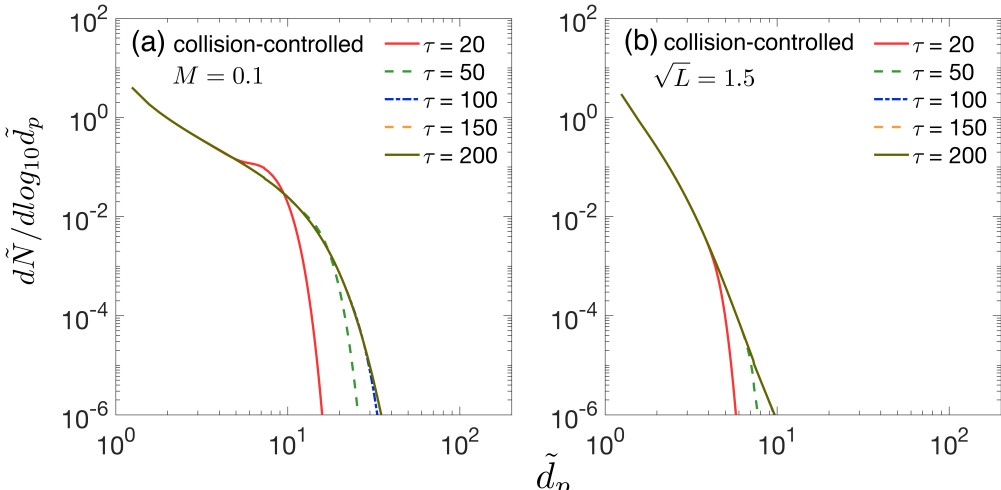

602

**Figure 6.** Particle size distribution at different dimensionless times for collision-controlled nucleation with **(a)** $M$=0.1
and **(b)** $\sqrt{L} = 1.5$. In both cases, sink processes not indicated in the figure were set to zero in the simulations. Particle
size distributions at certain times are not visible in the figure since they overlap with the particle size distribution at a
later time.



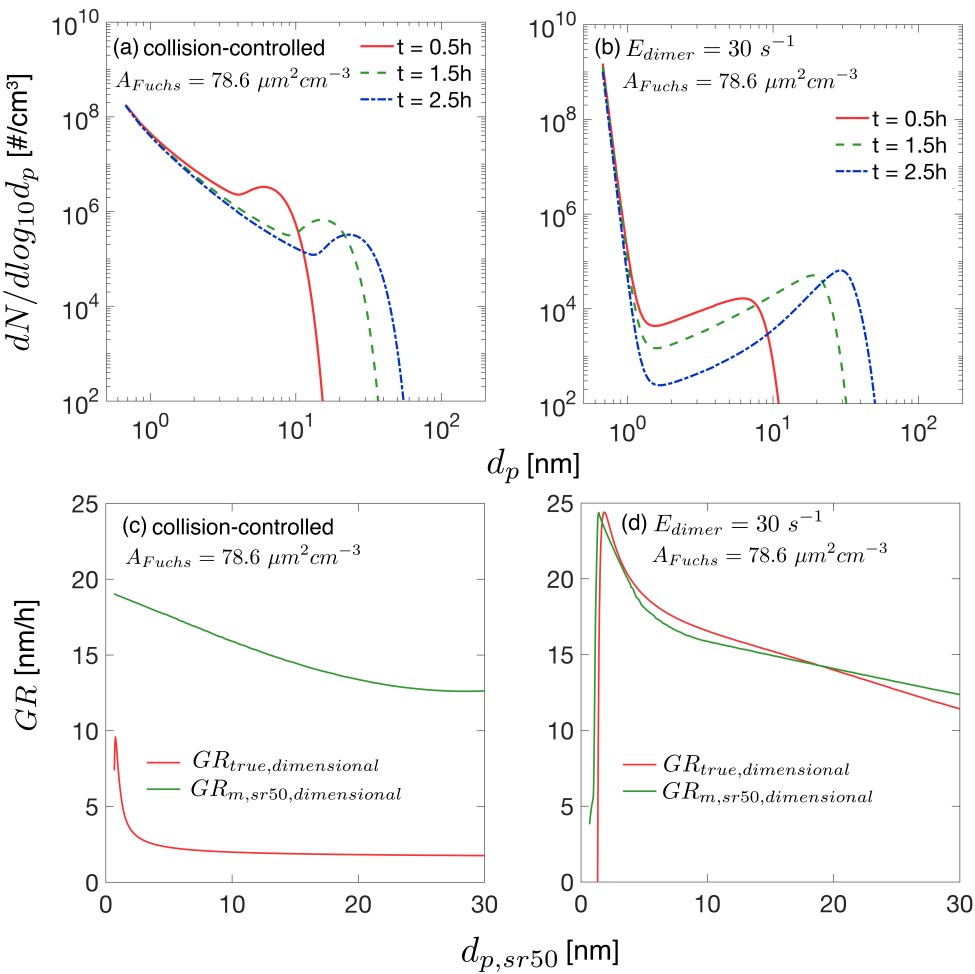


**Figure B1**. Dimensional particle size distribution and growth rates. The quantities shown in this figure are converted
from the dimensionless solution using Eqn. (6). The dimensional quantities involved in the conversions are $R =$
$1 \times 10^6$ cm$^{-3}$ s$^{-1}$, $\beta_{11\,fm} = 4.27 \times 10^{-10}$ cm$^3$ s$^{-1}$ and $v_1 = 1.62 \times 10^{-22}$ cm$^3$. The Fuchs surface area is 78.6
$\mu m^2\ cm^{-3}$, corresponding to $\sqrt{L}$=0.2. **(a)** Particle size distribution for collision controlled nucleation at $t = 0.5$h, 1.5h
and 2.5h. **(b)** Particle size distribution for nucleation with evaporation at $t = 0.5$h, 1.5h and 2.5h. Monomer evaporation
rate from dimer is 30 s$^{-1}$, corresponding to a dimensionless evaporation constant $E = 1 \times 10^{-3}$. **(c)** The dimensional
particle growth rates for collision-controlled nucleation as is shown in Fig. B1a. **(d)** The dimensional particle growth
rates for nucleation with evaporation as is shown in Fig. B1b.
