# Peer review of "Errors in Nanoparticle Growth Rates Inferred from"

_Atmospheric Chemistry and Physics, 2018_

## Referee Comment (RC1) · Anonymous Referee #1 · 18 Apr 2018

The present manuscript entitled "Errors in Nanoparticle Growth Rates Inferred from Measurements in Chemically Reacting Aerosol Systems" discusses the potential errors made by applying different methods to determine growth rates (in the following referred to as GR) from measured data. The respective approaches have in common that they do not distinguish between growth by vapor uptake and other processes that change the particle size distribution (e.g.: coagulation, dilution, wall loss or losses to preexisting particles). Thus the resulting error may, depending on the significance of those processes, be considerable. The authors analyze these errors qualitatively as well as quantitatively. They describe the effect of evaporation and pre-existing particles on the evolution of the (dimensionless) particle size distributions and as a consequence on

the growth rates. The application of non-dimensional quantities is elegant in this context and allows for applying the results to a variety of cases. Further, the manuscript is clearly written. As the methods described are still frequently used by scientists in recent publications (also high impact, such as PNAS, Science or Nature) for interpreting data, it is important to show their limitations and to point towards alternatives which are available.

General comments:

The authors focus on the contribution of particle-particle interaction to growth and determine a maximum error of the growth rate for the collision controlled scenario. They do not explicitly state that this error represents a maximum overestimation of the growth rate (there are several statements mentioning this "upper limit" of the GR [page 3, line 85; page 7 line 191; p 10, line 312] or "maximum possible error" [abstract]; however, it may be interpreted by the reader as the maximum value of the error). It may also be worth mentioning the possibility of GR underestimation caused by deposition losses, dilution and losses to pre-existing particles.

The effect of pre-existing particles on GR errors is discussed as a representative case for several processes (wall loss, dilution and pre-existing particles). This, according to the authors, is justified by findings on the similarity of those processes with regard to effects on the nucleation as described in a recent study (McMurry & Li, 2017). In the present manuscript it is assumed that particle sinks of any form mainly reduce the monomer concentration. Thus, the main effect is the reduction of nucleated particles and this limits coagulation which, according to the authors reduces the error in the GRs. However, loss of particles to the wall, to preexisting particles or by dilution is not considered by the analysis methods discussed and thus potentially lowers the GR obtained from the respective methods (e.g. in a case with low particle growth where uptake of vapor by the walls is limited while the walls may represent a perfect sink for particles). This results in underestimation of the GR.

In the manuscript errors of the analyzed GRs are discussed with regard to the analysis methods applied which are not suitable to produce size and time dependent GRs. The result of those methods is rather an array giving GR for various particle sizes and different measurement times. Further, the methods have inherent errors as they attribute any change of the PSD to growth. Thus, these methods in general are not suitable to produce realistic GR. However, in some specific cases they are. The present manuscript does not provide the necessary information to distinguish between situations where the methods can safely be applied or not. The reason is the fact that possible underestimation of the GR is not discussed (e.g. low GR in a chamber with considerable wall loss and/or dilution may lead to considerable underestimation of the GR by applying one of the methods used). Thus, I suggest removing statements on situations featuring safe usage of those methods and replacing them by statements indicating where the methods cannot/should not be applied. Maybe the authors should also point out once again the possible alternative methods for data analysis which do not suffer from the errors discussed in this manuscript in the conclusion section.

Specific:

p.2, line 40 (f): "Coagulation is accounted for with the coagulation integrals in the GDE and is a relatively well understood process that can be described with reasonable confidence in models." A reference would be helpful.

p.2, line 41 (f): "Growth involves processes that are not well understood for chemically complex aerosol systems, such as the atmosphere." Reference or examples plus references would be helpful.

p.4, line 95 (f): "Our results help to inform estimates of uncertainty for complex aerosol systems, such as the atmosphere, where errors are difficult to quantify." How is this possible as the present manuscript deals with nucleation of a single molecule species which is formed at a constant rate?

p.6, line 158: "and ðİŘÿk is the particle the evaporation rate". Remove the second "the"

p.7, line 190 (ff): "We believe collision-controlled nucleation (E=0) in the absence of other particle loss mechanisms such as wall deposition (W=0) and scavenging by pre-existing particles ( ðİŘ£=0) provides an upper limit to errors in GRm for a constant rate system (R=constant)." The error represents a maximum overestimation of the GR. A "maximum error" would also mean that it is bigger than the maximum underestimation of the GR which may not be true. Thus this statement is too general to me.

p.7, line 199: "Most noticeably, particles grow considerably faster at early stages of simulation" Do the particles really grow faster or do they seem to grow faster? What is the reason?

p.9, line 275: "Note for the range of ðİŘ£ values examined, the presence of preexisting particles alter GRtrue/GRm values by no more than 50%." The GRtrue/GRm ratio ranges from roughly 0.35 to about 1.1 which is more than 50% (see Fig. 4b)

p.10, line 306 (f): "In practice, this means measured growth rate based on all the four representative sizes can be a reasonable substitute of the true growth rate in a simi-lar nucleation scenario." As the possibility to underestimate the GR is not discussed, this statement does not hold true. Further, "similar nucleation scenario" is a vague statement. When would an experimental set of data be similar?

p.10. line 312: "Collision-controlled nucleation without preexisting particles results in an upper limit (up to a factor of 6) to discrepancies between true (GRtrue) and measured (e.g., GRm,mode) growth rates." It could be mentioned that this statement refers to simulated data (e.g.: Simulation showed that collision-controlled...) otherwise it is too general.

p.10, line 318 (f): "Both evaporation and preexisting particles bring GRtrue/ GRm closer to unity by decreasing the number of nucleated particles. In the case of evaporation, GRtrue/ GRm also increases as a result of elevated monomer concentration." This statement in general is not true. Evaporation and preexisting particles reduce the ratio GRtrue/GRm by reducing the overestimation caused by coagulation. In case the GR

is underestimated (i.e. GRtrue/GRm < 1; caused by e.g. wall losses/dilution combined with weak particle growth) by the analysis methods, the combined effect of evaporation and preexisting particles would even increase the error.

p.10 line 324 (f): "In this case, GRm based on all representative sizes can be a good approximation of GRtrue due to negligible coagulation effects." This statement, similar to the previous one, is too general as it considers only the possible overestimation of the GR (caused by coagulation). However, if the analysis method does not account for methods different from coagulation (e.g. dilution, wall loss, deposition), there may still be a significant difference between the measured and the "true" GR.

McMurry, P. H., & Li, C. (2017). The dynamic behavior of nucleating aerosols in constant reaction rate systems: Dimensional analysis and generic numerical solutions. Aerosol Science and Technology, 51(9), 1057-1070. doi: 10.1080/02786826.2017.1331292

---

## Referee Comment (RC2) · Anonymous Referee #2 · 23 Apr 2018

**Review of "Errors in Nanoparticle Growth Rates Inferred from Measurements in Chemically Reacting Aerosol Systems"**

**General comments**

In this study uncertainties in particle growth rates are investigated using model simulations. More specifically, the authors study how significantly the particle growth rates determined using different methods deviate from the growth rate due to vapor condensation. They show that this difference is largest in the system where the growth is collision–controlled and vapor concentrations are high, in which case the growth due to coagulation becomes significant. In the presence of sink due to pre-existing particles and evaporation, the coagulation growth is less significant and thus also the difference between the measured growth rate and the condensation growth rate is smaller.

The study seems scientifically sound and the presented results are interesting to the scientific community as the growth rate methods discussed in the manuscript are generally used when analyzing particle size distribution data. Therefore, I recommend the manuscript for publication in ACP after the authors have considered the comments listed below and the comments presented by Referee #1.

**Specific comments**

P1, L1: This study does not actually discuss the errors in nanoparticle growth rates but the difference between the measured growth rate and the growth rate caused by vapor condensation. These are separate issues because the growth due to collisions of small clusters (coagulation growth) is also real growth. Please modify the manuscript to make this clear (title, abstract, conclusions, and rest of the text).

P1, L18–20: It may be confusing for the reader to state that in the presence of pre-existing particles coagulation is reduced. You could make this clearer by writing, for example, "by reducing growth due to coagulation". The difference between coagulation losses of small particles due to pre-existing larger particles and coagulation growth caused by collisions of small clusters should be made clearer also elsewhere in the manuscript.

P2, L25: Instead of "growth", I would suggest writing here "condensation and evaporation" as all the other processes are also mentioned separately.

P2, L28: Removal of molecular species from a cluster cannot really be called "growth". Also, when discussing particle growth, it would be good to specify which size range is meant.

P2, L38–39: The difference between coagulation scavenging and the growth due to coagulation should be made clear also here. For example, writing "it is worthwhile to treat growth due to condensation and coagulation separately" would make this more understandable. In addition, although coagulation scavenging is rather well understood, the contribution of collisions of molecular clusters to the growth is not.

P2, L45: Please add references here for previous observations on GR.

P3, L56: GR is usually determined by linear fitting to diameter vs time data, instead of looking only at the difference between two sizes.

P3, L58: This method has also been applied in several studies for sub-3 nm particle size distribution data not measured by CPC batteries.

P3, L71: Please add a reference when discussing previous work. Also, this paragraph could fit better in the beginning of the introduction as it provides the general motivation of this work.

P3, L79: You should made it clear already here that you define $GR_{true}$ so that it is GR only due to vapor condensation.

P3, L82: $GR_{true}$ is defined in a different way by Kontkanen et al. (2016) and therefore using the same name for it is misleading.

P4, L103–116: The description of the model and model simulations could be slightly more detailed. The reader should understand the model without a need to look at the earlier publications.

P4, L119: Although using dimensionless parameters certainly has its benefits, it makes comparison between these results and experimental observations or previous simulations difficult. Therefore, also mentioning the values of corresponding dimensional variables (e.g. number concentration, diameter, GR, loss rate) for some of the key results (either in the text or in the figures) would be beneficial.

P7, L199: The fact that the particle growth rate due to condensation and evaporation is higher when there is evaporation in the system is difficult to understand.

P8, L230: Could you add a short explanation why different representative sizes follow this order?

P8, L238: Instead of referring to Eq. (6), could you explain the reason for higher GR?

P8, L242: This is now slightly unclear. Do you mean that the growth is first slow and then it accelerates?

P8, L245: What do you mean by using quotation marks with 'slow'?

P8, L248: Some of the measured GRs are in the beginning of the simulation lower than $GR_{true}$. This means that if evaporation rate was very high, the difference between $GR_m$ and $GR_{true}$ could possibly be larger than in the collision-limited case which is said to correspond to the case with "the maximum possible error".

P9, L253: But there seems to be even higher values at sizes lower than [10, 15]?

P9, L262: How does the coagulation sink depend on particle size in your simulations? When stating the range of $\sqrt{L}$ used in the simulations, it would be useful to mention the corresponding range for the dimensional variable.

P9, L274: This result sounds counterintuitive. Would the situation change if higher values of $\sqrt{L}$ were used? Does this situation correspond to the situation in the atmosphere? The collision-limited case probably occurs in the atmosphere in polluted environments where losses due to pre-existing particles are very high.

P11, L318–319: This conclusions is unclear as it is stated that $GR_{true}/GR_m$ both becomes closer to unity and increases due to evaporation.

**Technical comments**

P1, L18: Please add "that" after "show.

P6, L179: There is no need to repeat the name of the author twice.

P6, L182: Check the subscript.

P6, L188: Pleas add an en dash to show the range (also elsewhere).

P7, L215: Remove "of".

P8, L231: Please add "that" after "indicate".

P8, L248: Check the subscripts.

P9, L275: Please add "that" after "Note".

Figure 1: Please also mention what $d_{p,min}$ stands for in the figure caption.

---

## Author Comment (AC1) · 26 May 2018

The authors would like to thank the reviewers for their thoughtful reviews, and constructive comments and suggestions. Our replies are given directly after the comments (in bold); text that has been added/revised is shown in red font.

**General comments:**

**In this study uncertainties in particle growth rates are investigated using model simulations. More specifically, the authors study how significantly the particle growth rates determined using different methods deviate from the growth rate due to vapor condensation. They show that this difference is largest in the system where the growth is collision–controlled and vapor concentrations are high, in which case the growth due to coagulation becomes significant. In the presence of sink due to pre-existing particles and evaporation, the coagulation growth is less significant and thus also the difference between the measured growth rate and the condensation growth rate is smaller.**

**The study seems scientifically sound and the presented results are interesting to the scientific community as the growth rate methods discussed in the manuscript are generally used when analyzing particle size distribution data. Therefore, I recommend the manuscript for publication in ACP after the authors have considered the comments listed below and the comments presented by Referee #1.**

**Specific comments:**

**P1, L1: This study does not actually discuss the errors in nanoparticle growth rates but the difference between the measured growth rate and the growth rate caused by vapor condensation. These are separate issues because the growth due to collisions of small clusters (coagulation growth) is also real growth. Please modify the manuscript to make this clear (title, abstract, conclusions, and rest of the text).**

The reviewer correctly argues that collisions of small clusters can contribute to growth. We showed that the effect of those coagulation processes on particle growth rates ($GR$) can be significant for collision-controlled nucleation (Fig. 2a) but are much less important when cluster evaporation occurs to a significant extent (Fig. 2b). The reviewer argues that we should include growth due to cluster coagulation in our definition of "true" particle growth rates, $GR_{true}$. While there is some logic to this argument, we believe there is an even stronger argument to exclude growth due to cluster coagulation in the $GR$ (which we later define as the "true" growth rate $GR_{true}$) and $dd_p/dt$ terms defined by E1. 1 and 2 in our manuscript. Our argument might be viewed as merely semantic, but we believe it is more fundamental than this.

First, we acknowledge without question that the discovery of Lehtipalo and coworkers (2016), that clusters can contribute significantly to particle growth rates was a very significant discovery. It is important to understand all processes that contribute to growth, and this was the first paper to show explicitly that cluster coagulation is a significant contributor.

However, as the aerosol general dynamic equation has been formulated for several decades, cluster coagulation is explicitly included in the coagulation terms of the GDE and not in growth rate expression. This does not suggest that quantifying the contribution of cluster to growth is easy. Indeed, it is only recently that cluster distributions could be measured with sufficient accuracy to quantify this effect, and it is not done routinely in most studies. However, once these distributions are known, their dynamic behavior is logically included in the coagulation terms of the cluster balance equations. This allows one to account for the contributions of clusters to particle growth, as well as cluster-cluster coagulation for smaller particles, which can also be significant (Kurten 2018).

Because the reviewer refers to cluster coagulation as a growth process, we believe s/he would agree that it is described by the coagulation terms of the GDE. If so, however, it cannot also be included in the growth

term of the GDE, which applies to the net rate of particle growth due to molecular uptake (including condensation, evaporation, and other heterogeneous processes).  In addition to the mathematical arguments for not including cluster condensation as part of the growth term in the GDE, there are also conceptual arguments. If the cluster distribution is measured with sufficient accuracy to allow the effects of cluster coagulation on $GR_m$ to be quantified, that is a major step towards reconciling $GR_m$ with processes known to contribute towards particle growth. If large discrepancies remain after accounting for condensation, evaporation and cluster coagulation, that would underscore the need to study other types of processes that could also contribute (e.g., heterogeneous chemical reactions on or within particles.) Such heterogeneous processes are not understood, and the extent to which they may contribute to growth needs to be quantified.

Accordingly, we have not revised the manuscript to include cluster coagulation as a process that is included in our expression for "$GR_{true}$". We have chosen to conform to the original definition of growth by Friedlander, Seinfeld and their colleagues, and to only include molecular uptake for this term while acknowledging and quantifying the extent to which cluster coagulation can also contribute to growth.

**P1, L18–20: It may be confusing for the reader to state that in the presence of pre-existing particles coagulation is reduced. You could make this clearer by writing, for example, "by reducing growth due to coagulation". The difference between coagulation losses of small particles due to pre-existing larger particles and coagulation growth caused by collisions of small clusters should be made clearer also elsewhere in the manuscript.**

To be more specific about what coagulation is referred to, the sentence now reads 'This can lead to decreased discrepancies between measured growth rate and condensational growth rate by reducing coagulation between nucleated particles.'

**P2, L25: Instead of "growth", I would suggest writing here "condensation and evaporation" as all the other processes are also mentioned separately.**

"Growth" in this introductory sentence refers to all processes that lead to particle growth by molecular uptake. The subsequent paragraphs explain that these processes include condensation and evaporation, acid-base reactions, accretion, liquid phase chemical reactions, etc. Therefore, growth is not synonymous with condensation and evaporation in this context. We later explain that in this paper, the only growth processes that we include in this analysis are condensation and evaporation. However, it would be misleading to imply in the introduction that those are the only possible growth processes in general.

**P2, L28: Removal of molecular species from a cluster cannot really be called "growth". Also, when discussing particle growth, it would be good to specify which size range is meant.**

We give the definition of 'growth' here as net particle size change to addition or removal of molecular species. The sentence now reads 'Following established conventions long used in modeling aerosol dynamics (Friedlander, 2000;Gelbard and Seinfeld, 1979, 1980), we define the particle "growth rate" as the net rate of change in diameter of individual particles due to the addition or removal of molecular species. (If evaporation exceeds addition, the growth rate would be negative.)'

The result presented in this paper is germane to particle growth up to around 40 nm This information is given in the second to last paragraph in the introduction.

**P2, L38–39: The difference between coagulation scavenging and the growth due to coagulation should be made clear also here. For example, writing "it is worthwhile to treat growth due to condensation and coagulation separately" would make this more understandable. In addition, although coagulation scavenging is rather well understood, the contribution of collisions of molecular clusters to the growth is not.**

We agree that when interpreting experimental data, it would only be possible to account for all coagulation processes if the entire number distribution down to and including clusters of size 2 were accurately measured. However, if such data are available, contributions of coagulation to $GR_m$, can be accurately assessed. This is true for both coagulation scavenging and coagulation of the freshly nucleated particles. Because we understand our simulated data perfectly, we know those number distributions and can accurately calculate the effects of all coagulation interactions on $GR_m$. We clearly define growth as due to only to the net rate of molecular uptake (excluding all coagulation processes), thereby distinguishing between $GR_{true}$ and $GR_m$. We have added the following sentence to clarify this:

'The extent to which the coagulation of freshly nucleated molecular clusters contributes to measured growth rates can be accurately determined only if the entire number distribution down to clusters of size 2 is accurately measured. In the absence of such data, the contributions of cluster coagulation to growth could erroneously be attributed to vapor uptake.'

**P2, L45: Please add references here for previous observations on GR.**

We included Stolzenburg et al. (2005), Wang et al. (2013), Riccobono et al.(2012) and Tröstl et al.(2016) as references.

**P3, L56: GR is usually determined by linear fitting to diameter vs time data, instead of looking only at the difference between two sizes.**

Agreed. The sentence now reads 'The growth rate is obtained by first fitting a linear function of particle diameter (corresponding to the size bins) vs. time, and then calculating the slope of the fitted function'.

**P3, L58: This method has also been applied in several studies for sub-3 nm particle size distribution data not measured by CPC batteries.**

Agreed. The sentence now reads 'This approach has been used to analyze data from condensation particle counter (CPC) batteries (Riccobono, 2014), particle size magnifier (PSM) (Lehtipalo 2014), etc.'.

**P3, L71: Please add a reference when discussing previous work. Also, this paragraph could fit better in the beginning of the introduction as it provides the general motivation of this work.**

We have included the following references: Kontkanen et al.(2016), Riipinen et al.(2012), Hodshire et al.(2016), Smith et al. (2010), Smith et al. (2008) and Tröstl et al.(2016).

We agree that this paragraph fits better at the beginning of the introduction. Therefore, we combined this paragraph with the first paragraph of the introduction.

**P3, L79: You should make it clear already here that you define GR$_{true}$ so that it is GR only due to vapor condensation.**

Please refer to the response to the first specific comment.

**P3, L82: GR$_{true}$ is defined in a different way by Kontkanen et al. (2016) and therefore using the same name for it is misleading.**

To make clear the difference between $GR_{true}$ and the related concepts used by Kontkanen et al. (2016), the line now reads 'For example, Kontkanen (2016) used simulations to show that discrepancies between measured growth rate based on appearance time (AGR) and growth rate based on irreversible vapor condensation (CGR) can be significant. (Note $GR_{true}$ used in this paper differs from CGR in that $GR_{true}$ also incorporates evaporation.)'

**P4, L103–116: The description of the model and model simulations could be slightly more detailed. The reader should understand the model without a need to look at the earlier publications.**

Since loss to pre-existing particles and dilution are discussed in the manuscript, we added the definition of $\sqrt{L}$ and $M$ in the text (Eq. (3) and Eq. (4) in the revised manuscript). In addition, to better explain the model, we added the following text before introducing Eq.(6): 'The solution to the GDE for a constant rate system ($R$=constant) depends on dimensionless time, cluster size and the dimensionless variables $\sqrt{L}$, $M$, $E$, $\Omega$, etc., but is independent of the rate at which condensing vapor is produced by chemical reaction. That rate is required to transform the computed nondimensional solutions to dimensional results using simple multiplicative expressions given by McMurry and Li (2017):'.

In order for the reader to thoroughly understand what is discussed in the paper under review, she or he will need to read McMurry and Li (2017).

**P4, L119: Although using dimensionless parameters certainly has its benefits, it makes comparison between these results and experimental observations or previous simulations difficult. Therefore, also mentioning the values of corresponding dimensional variables (e.g. number concentration, diameter, GR, loss rate) for some of the key results (either in the text or in the figures) would be beneficial.**

This is a good suggestion. To facilitate comparison between dimensionless and dimensional results, we converted selected cases discussed in section 3.3 with assumed monomer production rates. The converted dimensional results are shown in Appendix B and Fig. B1.

**P7, L199: The fact that the particle growth rate due to condensation and evaporation is higher when there is evaporation in the system is difficult to understand.**

To better explain this we added the following sentence: 'Most noticeably, particles grow considerably faster at early stages of simulation. This occurs because evaporation depletes clusters and correspondingly increases monomer concentration. In the absence of pre-existing particles, monomer concentration accumulates until the supersaturation is high enough for nucleation to take place (see figure 2c). The accumulated monomers then rapidly condense on the nucleated particles, leading to rapid particle growth shown in Fig. 2b.'

**P8, L230: Could you add a short explanation why different representative sizes follow this order?**

This is an empirical result specific to the nucleation scenario discussed in this paper. We are not sure if this applies to all nucleation scenarios. As a result, we chose not to speculate as to whether or not this order might be a general result for all growth scenarios.

**P8, L238: Instead of referring to Eq. (6), could you explain the reason for higher GR?**

Some explanation is added to the end of the sentence. 'This is partly due to higher monomer concentrations (see red solid curve in Fig. 2c) and partly due to Eq. (6) that leads to higher true growth rate for smaller particles: the addition of a monomer leads to a bigger absolute as well as fractional diameter growth for small particles.'

**P8, L242: This is now slightly unclear. Do you mean that the growth is first slow and then it accelerates?**

Yes, the clusters containing a few monomers grow slowly due to the strong Kelvin effect. And particle growth then accelerates when the nucleation burst takes place. To make the text clearer, the paragraph is partially rewritten as follows: 'Figure 3d-3f are counterparts of Fig. 3a-3c, but with evaporation constant $E$ set to $1 \times 10^{-3}$. Figure 3d shows that $\tilde{d}_{p,sr50}$ and $\tilde{d}_{p,tot50}$ increase relatively slowly at the start of the simulation (see the amplified figure at the lower right corner of Fig. 3d; for reference, the dimensionless sizes of monomer, dimer and trimer are 1.24, 1.56 and 1.79 respectively). Subsequently, a marked change slope of the $\tilde{d}_p = \tilde{d}_p(\tau)$ curve is observed, indicating accelerated particle growth. This reflects that nucleation occurs with a burst of particle formation following a process of monomer and cluster

accumulation. The slow growth of the smallest clusters is an indication that the accumulation process is slow due to the strength of the Kelvin effect.'

**P8, L245: What do you mean by using quotation marks with 'slow'?**

The quotation mark has been deleted.

**P8, L248: Some of the measured GRs are in the beginning of the simulation lower than GR$_{true}$. This means that if evaporation rate was very high, the difference between GR$_m$ and GR$_{true}$ could possibly be larger than in the collision-limited case which is said to correspond to the case with "the maximum possible error".**

This is also pointed out by the other referee: $GR_m$ can be lower than $GR_{true}$. We didn't quantify underestimation of $GR_{true}$ by $GR_m$ in our revised manuscript. Therefore, to be more precise, we changed 'maximum possible error' to 'maximum overestimation of $GR_{true}$' wherever this is necessary.

**P9, L253: But there seems to be even higher values at sizes lower than [10, 15]?**

[10, 15] has been changed to be [5, 11].

**P9, L262: How does the coagulation sink depend on particle size in your simulations? When stating the range of $\sqrt{L}$ used in the simulations, it would be useful to mention the corresponding range for the dimensional variable.**

The dependence of loss rate to preexisting particles is $\sqrt{L}/k^{1/2}$, where $k$ is the number of monomer in a particle. This information is now given in the revised manuscript after Eq. (3) is introduced.

**P9, L274: This result sounds counterintuitive. Would the situation change if higher values of $\sqrt{L}$ were used? Does this situation correspond to the situation in the atmosphere? The collision-limited case probably occurs in the atmosphere in polluted environments where losses due to pre-existing particles are very high.**

We varied $\sqrt{L}$ values from 0 to 1 (results are not shown in the manuscript), and the monomer concentration varied by less than 10% for collision controlled simulation, though the number of nucleated particles decreased significantly. Dimensionally, if the monomer production rate is $R = 1\times 10^6$ cm$^{-3}$ s$^{-1}$, the monomer has a volume of $1.62\times10^{-22}$ cm$^3$ with a density of 1.47 g cm$^{-3}$ and the monomer collision frequency function $\beta_{11\,fm}$ is $4.27\times10^{-10}$ cm$^3$ s$^{-1}$, $\sqrt{L}$=1 corresponds to a Fuchs surface area $A_{Fuchs}$= 392 μm$^2$ cm$^{-3}$. This surface area is on the higher end of those observed in the atmosphere (Kuang et al. ,2010). Therefore, the results presented here are relevant to the atmosphere.

**P11, L318–319: This conclusion is unclear as it is stated that $GR_{true}$/$GR_m$ both becomes closer to unity and increases due to evaporation.**

Taking into account the possibility of underestimation $GR_{true}$ by $GR_m$, this conclusion now reads 'Both evaporation and scavenging by preexisting particles can reduce the concentration of particles formed by nucleation. Lower particle concentrations reduce the effect of coagulation on $GR_m$, so overestimation of $GR_{true}$ by $GR_m$ is lower than is found in the absence of these processes'.

**Technical comments**

**P1, L18: Please add "that" after "show.**

**P6, L179: There is no need to repeat the name of the author twice.**

**P6, L182: Check the subscript.**

**P6, L188: Pleas add an en dash to show the range (also elsewhere).**

**P7, L215: Remove "of".**

**P8, L231: Please add "that" after "indicate".**

**P8, L248: Check the subscripts.**

**P9, L275: Please add "that" after "Note".**

**Figure 1: Please also mention what *dp,min* stands for in the figure caption.**

The manuscript has been revised according to the referee's technical comments.

[revised manuscript text omitted]

---

## Author Comment (AC2) · 26 May 2018

The authors would like to thank the reviewers for their thoughtful reviews, and constructive comments and suggestions. Our replies are given directly after the comments (in bold); text that has been added/revised is shown in red font.

**General comments:**

**The authors focus on the contribution of particle-particle interaction to growth and determine a maximum error of the growth rate for the collision controlled scenario. They do not explicitly state that this error represents a maximum overestimation of the growth rate (there are several statements mentioning this "upper limit" of the GR [page 3, line 85; page 7 line 191; p 10, line 312] or "maximum possible error" [abstract]; however, it may be interpreted by the reader as the maximum value of the error). It may also be worth mentioning the possibility of GR underestimation caused by deposition losses, dilution and losses to pre-existing particles.**

**The effect of pre-existing particles on GR errors is discussed as a representative case for several processes (wall loss, dilution and pre-existing particles). This, according to the authors, is justified by findings on the similarity of those processes with regard to effects on the nucleation as described in a recent study (McMurry & Li, 2017). In the present manuscript it is assumed that particle sinks of any form mainly reduce the monomer concentration. Thus, the main effect is the reduction of nucleated particles and this limits coagulation which, according to the authors reduces the error in the GRs. However, loss of particles to the wall, to preexisting particles or by dilution is not considered by the analysis methods discussed and thus potentially lowers the GR obtained from the respective methods (e.g. in a case with low particle growth where uptake of vapor by the walls is limited while the walls may represent a perfect sink for particles). This results in underestimation of the GR.**

**In the manuscript errors of the analyzed GRs are discussed with regard to the analysis methods applied which are not suitable to produce size and time dependent GRs. The result of those methods is rather an array giving GR for various particle sizes and different measurement times. Further, the methods have inherent errors as they attribute any change of the PSD to growth. Thus, these methods in general are not suitable to produce realistic GR. However, in some specific cases they are. The present manuscript does not provide the necessary information to distinguish between situations where the methods can safely be applied or not. The reason is the fact that possible underestimation of the GR is not discussed (e.g. low GR in a chamber with considerable wall loss and/or dilution may lead to considerable underestimation of the GR by applying one of the methods used). Thus, I suggest removing statements on situations featuring safe usage of those methods and replacing them by statements indicating where the methods cannot/should not be applied. Maybe the authors should also point out once again the possible alternative methods for data analysis which do not suffer from the errors discussed in this manuscript in the conclusion section.**

**Reply to general comments:**

We find the review very constructive and have improved our paper accordingly. Major changes include

1. We added Sect. 3.4 and Fig. 6 in the revised manuscript to qualitatively show that in the presence of strong particle sinks, true growth rate can be underestimated by measured growth rate. In such nucleation scenarios, the particle size distribution approaches steady state after a certain time with the measured growth rate approaching 0, but the true growth rate remains finite and is thus underestimated by measured growth rate.
2. Since we do not study underestimation of growth quantitatively, we changed 'maximum possible error' or similar expressions to 'maximum overestimation of $GR_{true}$ by $GR_m$' or similar expressions throughout the manuscript.

3. Statements regarding safe usage of using measured growth rate as true growth rates have been removed; instead, we mainly focus on the discussing the simulation results presented in the paper and avoid making overly general statements.

**Reply to specific comments:**

**p.2, line 40 (f): "Coagulation is accounted for with the coagulation integrals in the GDE and is a relatively well understood process that can be described with reasonable confidence in models." A reference would be helpful**

We included Chan and Mozurkewich (2001) and Kürten et al. (2018). In the former reference coagulation rates were measured experimentally and Hamaker constant were otained by fitting experimental data. The result were then applied in the latter reference to analyze CLOUD data.

**p.2, line 41 (f): "Growth involves processes that are not well understood for chemically complex aerosol systems, such as the atmosphere." Reference or examples plus references would be helpful.**

We included Barsanti et al. (2009), Riipinen et al. (2012) and Hodshire et al. (2016) as references.

**p.4, line 95 (f): "Our results help to inform estimates of uncertainty for complex aerosol systems, such as the atmosphere, where errors are difficult to quantify." How is this possible as the present manuscript deals with nucleation of a single molecule species which is formed at a constant rate?**

We think our original statement is a bit overreaching. The corresponding text now reads "Our results help to inform estimates of uncertainties for systems with a single nucleating species, or systems that can be modeled in a similar way to a single species system (Kürten et al. ,2018)."

**p.6, line 158: "and $E_k$ is the particle the evaporation rate". Remove the second "the".**

'The' has been removed.

**p.7, line 190 (ff): "We believe collision-controlled nucleation (E=0) in the absence of other particle loss mechanisms such as wall deposition (W=0) and scavenging by preexisting particles ( $\sqrt{L}$=0) provides an upper limit to errors in GRm for a constant rate system (R=constant)." The error represents a maximum overestimation of the GR. A "maximum error" would also mean that it is bigger than the maximum underestimation of the GR which may not be true. Thus this statement is too general to me.**

Agreed. We reworded the sentence to be "We believe collision-controlled nucleation ($E$=0) in the absence of other particle loss mechanisms such as wall deposition ($W$=0) and scavenging by preexisting particles ( $\sqrt{L}$=0) provides an upper limit for overestimation of $GR_{true}$ for a constant rate system ($R$=constant)."

**p.7, line 199: "Most noticeably, particles grow considerably faster at early stages of simulation" Do the particles really grow faster or do they seem to grow faster? What is the reason?**

The following sentences were added to explain the faster particle growth at the early stage of simulation: "This occurs because evaporation depletes clusters and correspondingly increases monomer concentration. In the absence of pre-existing particles, monomer concentration accumulates until the supersaturation is high enough for nucleation to take place (see figure 2c). The accumulated monomers then rapidly condense on the nucleated particles, leading to the rapid particle growth shown in figure 2b."

**p.9, line 275: "Note for the range of $\sqrt{L}$ values examined, the presence of preexisting particles alter $GR_{true}/GR_m$ values by no more than 50%." The GR$_{true}$/GR$_m$ ratio ranges from roughly 0.35 to about 1.1 which is more than 50% (see Fig. 4b)**

The original text "Note for the range of $\sqrt{L}$ values examined, the presence of preexisting particles alter $GR_{true}/GR_m$ values by no more than 50%" is a comment on collision-controlled nucleation ($E=0$). Fig. 4b shows the difference between each curve (corresponding to different $\sqrt{L}$ values) is indeed less than 50%. To avoid confusion, "for collision controlled nucleation" is added to the original text.

**p.10, line 306 (f): "In practice, this means measured growth rate based on all the four representative sizes can be a reasonable substitute of the true growth rate in a similar nucleation scenario." As the possibility to underestimate the GR is not discussed, this statement does not hold true. Further, "similar nucleation scenario" is a vague statement. When would an experimental set of data be similar?**

This sentence has been deleted and the analysis in the revised manuscript is focused only on the simulation results.

**p.10. line 312: "Collision-controlled nucleation without preexisting particles results in an upper limit (up to a factor of 6) to discrepancies between true ($GR_{true}$) and measured (e.g., $GR_{m,mode}$) growth rates." It could be mentioned that this statement refers to simulated data (e.g.: Simulation showed that collision-controlled) otherwise it is too general.**

Agreed. The sentence in question now reads "Simulated data shows that collision-controlled nucleation without pre-existing particles leads to an upper limit (up to a factor of 6) of overestimating true growth rates ($GR_{true}$) by modal growth rates ($GR_{m,mode}$)."

**p.10, line 318 (f): "Both evaporation and preexisting particles bring $GR_{true}$/ $GR_m$ closer to unity by decreasing the number of nucleated particles. In the case of evaporation, $GR_{true}$/ $GR_m$ also increases as a result of elevated monomer concentration." This statement in general is not true. Evaporation and preexisting particles reduce the ratio $GR_{true}/GR_m$ by reducing the overestimation caused by coagulation. In case the GR is underestimated (i.e. $GR_{true}/GR_m < 1$; caused by e.g. wall losses/dilution combined with weak particle growth) by the analysis methods, the combined effect of evaporation and preexisting particles would even increase the error**

The sentence now reads "Both evaporation and scavenging by preexisting particles can reduce the concentration of particles formed by nucleation. Lower particle concentrations reduce the effect of coagulation on $GR_m$, so overestimation of $GR_{true}$ by $GR_m$ is lower than is found in the absence of these processes". In addition, we added section 3.4 to briefly discuss the situation where strong particle sink processes (i.e., sufficiently large values of $M$ or $\sqrt{L}$) lead to steady state particle size distributions. In these cases, measurements would not reveal any particle growth after a certain time and $GR_m$ would approach 0.

**p.10 line 324 (f): "In this case, $GR_m$ based on all representative sizes can be a good approximation of $GR_{true}$ due to negligible coagulation effects." This statement, similar to the previous one, is too general as it considers only the possible overestimation of the GR (caused by coagulation). However, if the analysis method does not account for methods different from coagulation (e.g. dilution, wall loss, deposition), there may still be a significant difference between the measured and the "true" GR.**

This statement has been deleted since it is too general.

Barsanti, K. C., McMurry, P. H., and Smith, J. N.: The potential contribution of organic salts to new particle growth, Atmos. Chem. Phys., 9, 2949-2957, 10.5194/acp-9-2949-2009, 2009.

Chan, T. W., and Mozurkewich, M.: Measurement of the coagulation rate constant for sulfuric acid particles as a function of particle size using tandem differential mobility analysis, Journal of Aerosol Science, 32, 321-339, https://doi.org/10.1016/S0021-8502(00)00081-1, 2001.

Hodshire, A. L., Lawler, M. J., Zhao, J., Ortega, J., Jen, C., Yli-Juuti, T., Brewer, J. F., Kodros, J. K., Barsanti, K. C., Hanson, D. R., McMurry, P. H., Smith, J. N., and Pierce, J. R.: Multiple new-particle growth pathways observed at the US DOE Southern Great Plains field site, Atmos. Chem. Phys., 16, 9321-9348, 10.5194/acp-16-9321-2016, 2016.

Kürten, A., Li, C., Bianchi, F., Curtius, J., Dias, A., Donahue, N. M., Duplissy, J., Flagan, R. C., Hakala, J., Jokinen, T., Kirkby, J., Kulmala, M., Laaksonen, A., Lehtipalo, K., Makhmutov, V., Onnela, A., Rissanen, M. P., Simon, M., Sipilä, M., Stozhkov, Y., Tröstl, J., Ye, P., and McMurry, P. H.: New particle formation in the sulfuric acid–dimethylamine–water system: reevaluation of CLOUD chamber measurements and comparison to an aerosol nucleation and growth model, Atmos. Chem. Phys., 18, 845-863, 10.5194/acp-18-845-2018, 2018.

Riipinen, I., Yli-Juuti, T., Pierce, J. R., Petäjä, T., Worsnop, D. R., Kulmala, M., and Donahue, N. M.: The contribution of organics to atmospheric nanoparticle growth, Nature Geoscience, 5, 453, 10.1038/ngeo1499, 2012.